

# Simulation of the radiative effect of haze on urban hydrological cycle using reanalysis data in Beijing

Tom V. Kokkonen[1], Susan B. Grimmond[2], Sonja Murto[1,3], Huizhi Liu[4], Anu-Maija Sundström[5], and Leena Järvi[1,6]

[1]Institute for Atmospheric and Earth System Research / Physics, Faculty of Science, University of Helsinki, Finland
[2]Department of Meteorology, University of Reading, UK
[3]Department of Meteorology, University of Stockholm, Sweden
[4]Institute of Atmospheric Physics, Chinese Academy of Sciences, Beijing, China
[5]Earth Observation, Finnish Meteorological Institute, Helsinki, Finland
[6]Helsinki Institute of Sustainability Science, University of Helsinki, Finland

**Correspondence:** Tom Kokkonen (tom.kokkonen@helsinki.fi), Huizhi Liu (huizhil@mail.iap.ac.cn)

**Abstract.** Although, air pollution modifies local air temperatures and boundary layer structure in urban areas, little is known about its effects on the urban hydrological cycle. To explore the radiative effect of haze on changes in the urban surface water balance during different haze levels are modelled in Beijing using the Surface Urban Energy and Water Balance Scheme (SUEWS), forced by reanalysis data. The pollution levels are classified using aerosol optical depth observations. We show how

the reanalysis radiation data do not include the attenuating effect of haze and develop a haze correction for the incoming solar radiation. With this haze correction the SUEWS model simulates the eddy covariance measured latent heat flux well.

Both surface runoff and drainage increase with severe haze levels particularly with low precipitation rates: runoff from 0.06 to 0.18 mm day$^{-1}$ and drainage from 0.43 to 0.62 mm day$^{-1}$ during fairly clean and extremely polluted conditions, respectively. When all precipitation events are taken into account, runoff is higher during the extremely polluted conditions than with cleaner

conditions except during the cleanest conditions when the high precipitation rates induces largest runoff. Thus, the radiative effect of haze is not likely impacting on the likelihood of flash floods. However, the low runoff rates commonly transport pollutants to soil and water and therefore their changes are important to understanding detailed deterioration of urban soil and aquatic environments.

**1  Introduction**

In recent decades rapid economic development and acceleration of urbanization and industrialization has led to many environmental problems in China, such as atmospheric and water pollution, and soil contamination (e.g., Kulmala, 2015; Liao et al., 2015; Shao et al., 2006; Sun et al., 2014; Xia et al., 2011). As a consequence of urbanization and industrialization, northeast China is one of the most populated and most polluted areas in the world (HEI International Scientific Oversight Committee,



2010). With the continuing growth of urban population, serious water shortages, deterioration of water quality and aquatic environment are becoming concerns for municipal hydrological authorities (Li et al., 2015; Sun et al., 2014). As urbanization increases the extent of impervious surfaces, enhancing the likelihood of surface floods from surface runoff (e.g., Rodriguez et al., 2003), understanding the local urban hydrological cycle, potential increase in surface flooding and pollutant loads to

5 urban water bodies are important.

Atmospheric pollution modifies local urban climate by decreasing solar radiation at the surface further lowering the near surface air temperatures (Ding et al., 2013; Wang et al., 2014), turbulent heat fluxes and boundary layer heights and increases incoming longwave radiation emitted by the polluted layer (Miao et al., 2009; Petäjä et al., 2016; Tang et al., 2016). However, the linkage between atmospheric pollution and urban hydrological cycle has not yet been studied despite its potential contribu-

10 tion to deterioration of urban water bodies. This may be because of lack of high resolution meteorological and/or hydrological observations or their availability needed for detailed analyses and modelling from regions with heavy air pollution episodes. Global reanalysis products could provide the essential variables to enable modelling where needed observations are unavailable or have too coarse temporal resolution (Kokkonen et al., 2018b). A number of reanalysis products are available but to our knowledge none have been properly evaluated in highly polluted urban environments.

The aims of this study are to (1) explore how atmospheric pollution modifies the local urban hydrological cycle and (2) to examine the quality of the reanalysis data in highly polluted Beijing, China. The evaluated reanalysis product is the WATCH Forcing Data ERA-Interim (WFDEI, Weedon et al., 2014) with the focus on the most important meteorological variables (precipitation, solar radiation, air temperature) controlling the hydrological cycle and modelling. In addition, the urban land surface model Surface Energy and Water Balance Scheme (SUEWS) (Järvi et al., 2011; Ward et al., 2017) used to simulate the

urban hydrological cycle in different haze levels is evaluated against eddy covariance measured latent and sensible heat fluxes. Aerosol optical depth observations are used to classify the pollution levels for the assessment of the impact of radiative effect of haze on the local urban water balance in Beijing for 2001–2013.

## 2  Hydrological modelling

The hydrological modelling is conducted using the Surface Urban Energy and Water Balance Scheme (SUEWS, Järvi et al.,

2011) version V2017b (Ward et al., 2017, 2018). SUEWS is an urban land surface model that simulates the surface energy and water balances at the local (neighborhood) scale. In SUEWS, the urban surface is separated into seven hydrologically connected surface types (buildings, paved surfaces, grass, evergreen trees/shrub, deciduous trees/shrubs and water) each having a single soil layer below, excluding the water surface. For each surface type, evaporation is calculated using the Penman-Monteith equation (Monteith, 1965; Penman, 1948) modified for urban environments (Grimmond and Oke, 1991), and runoff from a

running water balance (Grimmond and Oke, 1991; Järvi et al., 2011). SUEWS has been optimized to run with minimum amount of model forcing data and includes sub-models for net all-wave radiation, irrigation ($I$) and anthropogenic heat flux. The overall parameters used in the model runs are given in Table 1. The performance of SUEWS and sensitivity to input variables and parameterisation has been extensively evaluated in the past studies in different climates and for multiple variables


**Table 1.** Overall model parameter values used at model runs in Beijing. See Table A1 for notation and Järvi et al. (2011, 2014); Ward et al. (2016) for data sources.

| | | | | | | | | |
|---|---|---|---|---|---|---|---|---|
| $\alpha_s^{\min}$ | 0.18 | $a_t$ | 0.07 mm °C$^{-1}$ h$^{-1}$ | $G1$ | 3.5 | $R_C$ | 1.0 mm |
| $\alpha_s^{\max}$ | 0.85 | $a_{0,\{wd,we\}}$ | 0.308 W m$^{-2}$(p$^{-1}$ ha$^{-1}$)$^{-1}$ | $G2$ | 200 W m$^{-2}$ | $S_1$ | 5.56 |
| $\epsilon_s$ | 0.99 | $a_{1,\{wd,we\}}$ | 0.0099 W m$^{-2}$ K$^{-1}$ (p$^{-1}$ ha$^{-1}$)$^{-1}$ | $G3$ | 0.13 | $S_2$ | 0 mm |
| $\rho_e$ | 200 kg m$^{-3}$ | $a_{2,\{wd,we\}}$ | 0.0102 W m$^{-2}$ K$^{-1}$ (p$^{-1}$ ha$^{-1}$)$^{-1}$ | $G4$ | 0.7 | $S_{\mathrm{pipe}}$ | 100 mm |
| $\rho_s^{\min}$ | 100 kg m$^{-3}$ | $b_{0,a}$ | -19.19 mm | $G5$ | 30 °C | SDD | -450 |
| $\rho_s^{\max}$ | 400 kg m$^{-3}$ | $b_{1,a}$ | 2.22 mm K$^{-1}$ | $G6$ | 0.05 mm$^{-1}$ | $T_{air}^{\mathrm{initial}}$ | -2.7 °C |
| $\tau_a$ | 0.006 | $b_{2,a}$ | 0.78 mm d$^{-1}$ | GDD | 300 | $T_{\mathrm{BaseGDD}}$ | 5 °C |
| $\tau_f$ | 0.0367 | $b_{0,m}$ | -5.76 mm | $I_w$ | 0 mm | $T_{\mathrm{BaseSDD}}$ | 11 °C |
| $a_1$ | 0.25 | $b_{1,m}$ | 0.67 mm K$^{-1}$ | $K\!\downarrow_m$ | 1200 W m$^{-2}$ | $T_{\mathrm{BaseQF}}$ | 18.2 °C |
| $a_2$ | 0.6 | $b_{2,m}$ | 0.24 mm d$^{-1}$ | $K_s$ | 0.0005 mm s$^{-1}$ | $T_{\mathrm{lim}}$ | 2.2 °C |
| $a_3$ | -30 | $C_{\min}^R$ | 0.05 mm | $r_s^{\max}$ | 9999 s m$^{-1}$ | $T_H$ | 55 °C |
| $a_f$ | 1 | $C_{\max}^R$ | 0.2 mm | res$_{\mathrm{cap}}$ | 10 mm | $T_L$ | -10 °C |
| $a_r$ | 0.0016 mm W$^{-1}$ h$^{-1}$ | DaysSinceRain | 28 | res$_{\mathrm{drain}}$ | 0.25 mm h$^{-1}$ | $T_{\mathrm{step}}$ | 300 s |

(e.g., Alexander et al., 2015; Demuzere et al., 2017; Järvi et al., 2011, 2014, 2017; Karsisto et al., 2016; Kokkonen et al., 2018a, b; Ward et al., 2016, 2018).

The study area is a 1 km radius circle around the 325 m high Institute of Atmospheric Physics (IAP) meteorological measurement tower (39.97° N, 116.37° E) located in the north-western part of Beijing, China, at the Haidian district (Liu et al.,
2012). This circle approximates the source area of the eddy covariance (EC) measurements at height 47 m (Liu et al., 2012) used to evaluate SUEWS model performance. This area is densely built (70 % of impervious surfaces) urban area (Local Climate Zone (LCZ) 1; Stewart and Oke, 2012) with only 29 % of vegetated surfaces and 1 % of open water. The surface cover fractions for the study area are calculated from aerial photographs using GIS-software (ArcGIS 10.1) and digitalized using two freely available base maps: OpenStreetMap (OpenStreetMap contributors, 2015) and World Imagery (Esri, 2009), following
the methods in Murto (2017). With the available imagery, separation to evergreen and deciduous trees and shrubs needed by the model is not possible using GIS-methods. Therefore these fractions (15 % and 85 % of fraction of vegetation, respectively) together with mean tree height (8 m) are estimated based on the common tree species to be found in Haidian district (Ma and Liu, 2003). The mean building height is 19.1 m (Miao et al., 2012).

The population density is estimated from 1 km gridded population dataset for 2010 (Fu et al., 2014). The grid population
densities are weighted by their areal fractions within the study area. There has been no further urbanization at the study site (Cheng et al., 2018) and therefore population density and surface characteristics are assumed to stay constant throughout the study period.

The WFDEI (Weedon et al., 2014) meteorological forcing data are derived for hydrological modelling purposes from ERA-Interim (Dee et al., 2011) reanalysis product via sequential interpolation to half-degree resolution with 3 h temporal resolution.
Bias correction with quantile mapping (BCQM) is applied to downscale the daily precipitation totals (Kokkonen et al., 2018b). The 5 min time-step calculations are disaggregated in a nonlinear manner to provide realistic precipitation pattern from coarse





**Table 2.** Instruments used on the 325 m IAP tower (47 m level; Liu et al. (2012)).

| Physical quantity | Instrument | Model |
| --- | --- | --- |
| Three-dimensional wind velocity | Three-dimensional sonic anemometer | CSAT-3 |
| $H_2O$ density | Infrared gas analyser | LI-7500 |
| Incoming solar radiation | Radiometer | CNR1 |
| Temperature | Thermometer | Developed by the Institute of Atmospheric Physics |
| Humidity | Hygrometer | Developed by the Institute of Atmospheric Physics |
| Wind speed and direction | Cup anemometers and vanes | Developed by the Institute of Atmospheric Physics |

input data (Ward et al., 2018) (Table A3). The air temperature ($T_{air}$) and pressure are adjusted to simulation height using environmental lapse rate ($\Gamma = -6.5$ K km$^{-1}$) and the hypsometric equation (Kokkonen et al., 2018b; Weedon et al., 2010). The WFDEI data are downscaled from 3 h to 5 min temporal resolution of the model time-step within the model (Ward et al., 2017).

The WFDEI reanalysis data in Beijing are evaluated for 2006–2009 using observed meteorological variables, including hourly $T_{air}$, relative humidity (RH) and incoming solar radiation ($K\downarrow$) measured on the IAP tower at 47 m level (Liu et al., 2012) (Table 2) and daily precipitation ($P$) 10 km southwest of the tower (Menne et al., 2012a, b). The same years are used to evaluate SUEWS against the IAP EC measurements from the same 47 m level. The 47 m level of IAP tower is in the roughness sublayer for wind directions mainly from southwest and northwest (Miao et al., 2012). Therefore the wind directions with buildings over 50 m high (314–3°, 40–45°, 112–128°, 160–243°) are filtered out from the EC observations (34 % of the data).

SUEWS is run for 2000 to 2013, with the first year as a spin-up period, leaving years 2001–2013 for the analysis. The hydrological cycle is analysed during the thermal summer (Apr–Sep) as the main focus of the study is in water balance and the winter time in Beijing is extremely dry. For example in 2013 the precipitation occurred between Oct–Mar covers only 6 % (33.7 mm) of the annual precipitation (Beijing Municipal Bureau of Statistics, 2016). Due to difference in behaviour in summer and winter months, the two periods should be analysed separately and this would leave insufficient amount of data for statistical analysis in winter months. The polluted and non-polluted days in the studied years are separated based on aerosol optical depth (AOD, 440 nm) obtained from AERONET station (Che et al., 2009; Holben et al., 1998) located at the study site.

### 2.1 Statistical methods

The pollution levels are obtained by dividing the AOD observations from the whole study period (2001–2013) into four quantiles (i.e. roughly equal amount of data in all of the air quality classes), i.e. extremely polluted air (AOD>1), polluted air (0.438–1), low pollution (0.203–0.438) and fairly clean air (<0.203).

The hydrological analysis is made stratifying the results by different pollution levels described above and different percentiles of daily precipitation from the study period (2001–2013). The hydrological components are divided into four ranges of daily precipitation percentiles (0–25, 0–50, 0–75, 0–100) including dry days. The further statistical analysis of the results of $P$ (stratified already by different percentiles) includes only wet days, but the other variables analysed include also the dry days.





Box plots used to explore the hydrological cycle, give the median, and the interquartile range (IQR), with whiskers of 1.5 IQR. The box plots have notches which indicate the 95 % confidence levels.

The linear correlations among different variables are analysed using common statistical tools, including root mean square error (RMSE), RMSE normalised with standard deviation of observations (nRMSE), mean bias error (MBE), MBE normalised with mean of observations (nMBE), mean absolute error (MAE), MAE normalised with mean of observations (nMAE) and Pearson's correlation coefficient ($r$). The regression lines have been calculated for scatter plots after applying the Lowess smoothing (Cleveland, 1979, 1981). The performance of model runs and WFDEI variables are evaluated using a Taylor (2001) diagram.

## 3  Results

### 3.1  Evaluation of WFDEI data in polluted urban environment

Poor air quality is a result of several factors including pollutant emissions, atmospheric transport, atmospheric chemistry and meteorological conditions. Therefore the accuracy of meteorological variables in reanalysis products is essential to be able to study the effects of pollutants correctly to local hydrological cycle.

Although, extensive evaluation of WFDEI data have been undertaken (e.g., Weedon et al., 2014) polluted urban areas have so far been neglected. Here WFDEI $P$, $K\downarrow$, RH and $T_{air}$ (here after the subscript $WF$ indicates WFDEI variables), the most important input variables for hydrological modelling with SUEWS (Alexander et al., 2015; Kokkonen et al., 2018b; Ward and Grimmond, 2017) are evaluated.

Haze is known to attenuate $K\downarrow$ in highly polluted environments but this attenuation is not properly accounted for in the WFDEI data (Fig. 1 and 2, Table 3) because sometimes haze may be from local emissions and because of secondary nucleation. Overestimation of hourly $K\downarrow_{WF}$ against the observed values increases with the level of pollution (nMBE: -0.01, 0.00, 0.08, 0.28; fairly clean, low pollution, polluted and extremely polluted air, respectively; see Sect. 2.1 for details). Thus hourly $K\downarrow_{WF}$ is corrected using observations between 2006–2009 from the 325 m IAP measurement tower separately for thermal summer (Apr–Sept) and winter (Oct–Mar) due to slightly different behaviour (Fig. 1 and A1). First a Lowess smoothing is applied to observed $K\downarrow$ normalized with the clear sky radiation (determined from $I_{SC} \times \cos\theta_z$, where $I_{SC}$ is the solar constant (1367 W m$^{-2}$) and $\theta_z$ is the solar zenith angle) as a function of AOD. Second, regression coefficients for different times of day are determined (Fig. 1). Before corrections are applied to WFDEI data, the $K\downarrow_{WF}$ is downscaled from 3 h to 1 h temporal resolution (Kokkonen et al., 2018b). The corrections are made by fitting the hourly $K\downarrow_{WF}$ data for the whole study period (2001–2013) using regression coefficients when AOD observations are available ($N = 20462$). The developed correction increases substantially the $K\downarrow_{WF}$ accuracy during pollution events bringing the more polluted levels closer to the cleaner levels (nMBE: -0.01, 0.00, -0.03, -0.03 from clean to extremely polluted conditions).

The height corrected $T_{air,WF}$ (Kokkonen et al., 2018b) correlates with observations well ($r>0.93$) and the nMBE is low (up to 0.26; Table 3). The nMBE of RH$_{WF}$ is also low (from -0.11 to 0.11) and the correlation coefficient reasonably good ($>0.68$).





**Table 3.** Comparison of hourly WFDEI meteorological variables (except daily for $P$) 2006–2009 stratified by pollution levels (extremely polluted air (AOD>1), polluted air (0.438–1), low pollution (0.203–0.438), fairly clean air (<0.203) (see Sect. 2.1 for details). Superscript $uc$ indicates uncorrected variables. For explanation of the statistical methods see Sect. 2.1.

|  | Variable | Level of pollution | $N$ | $r$ | RMSE | nRMSE | MBE | nMBE | MAE | nMAE |
|---|---|---|---|---|---|---|---|---|---|---|
| WFDEI | $RH_{WF}$ | Extreme | 1557 | 0.72 | 13.73 | 0.82 | -5.44 | -0.11 | 10.83 | 0.23 |
|  |  | Polluted | 1748 | 0.72 | 13.00 | 0.75 | -2.36 | -0.06 | 9.90 | 0.26 |
|  |  | Low | 1523 | 0.74 | 11.45 | 0.70 | 0.23 | 0.01 | 8.34 | 0.28 |
|  |  | Clean | 1290 | 0.68 | 10.21 | 0.81 | 3.05 | 0.11 | 7.84 | 0.29 |
|  | $T_{air,WF}$ | Extreme | 1557 | 0.94 | 5.63 | 0.56 | 4.25 | 0.25 | 4.58 | 0.27 |
|  |  | Polluted | 1748 | 0.93 | 5.88 | 0.54 | 3.94 | 0.25 | 4.48 | 0.29 |
|  |  | Low | 1523 | 0.95 | 4.90 | 0.43 | 3.21 | 0.23 | 3.90 | 0.28 |
|  |  | Clean | 1290 | 0.97 | 4.47 | 0.37 | 2.96 | 0.26 | 3.65 | 0.33 |
|  | $P_{WF}$ | Extreme | 70 | 0.38 | 13.23 | 1.06 | -2.88 | -0.37 | 7.05 | 0.90 |
|  |  | Polluted | 69 | 0.68 | 16.20 | 1.33 | 3.10 | 0.42 | 8.39 | 1.14 |
|  |  | Low | 53 | 0.55 | 10.35 | 1.00 | -1.64 | -0.18 | 6.32 | 0.71 |
|  |  | Clean | 47 | 0.67 | 10.19 | 0.77 | -2.81 | -0.33 | 5.82 | 0.69 |
|  | $P_{WF}^{uc}$ | Extreme | 82 | 0.42 | 11.15 | 0.95 | -2.37 | -0.33 | 5.94 | 0.83 |
|  |  | Polluted | 80 | 0.73 | 9.78 | 0.85 | 1.34 | 0.20 | 6.10 | 0.92 |
|  |  | Low | 62 | 0.49 | 10.08 | 0.95 | -2.95 | -0.33 | 6.20 | 0.68 |
|  |  | Clean | 54 | 0.66 | 10.04 | 0.80 | -2.32 | -0.31 | 5.19 | 0.70 |
|  | $K{\downarrow}_{WF}$ | Extreme | 1557 | 0.94 | 70.08 | 0.33 | 8.63 | -0.03 | 44.46 | 0.13 |
|  |  | Polluted | 1748 | 0.96 | 68.86 | 0.27 | 13.23 | -0.03 | 40.98 | 0.10 |
|  |  | Low | 1523 | 0.96 | 72.83 | 0.27 | 0.57 | 0.00 | 40.82 | 0.09 |
|  |  | Clean | 1290 | 0.96 | 75.24 | 0.28 | -5.44 | -0.01 | 43.08 | 0.11 |
|  | $K{\downarrow}_{WF}^{uc}$ | Extreme | 1557 | 0.90 | 132.88 | 0.63 | 93.28 | 0.28 | 108.95 | 0.32 |
|  |  | Polluted | 1748 | 0.93 | 98.08 | 0.39 | 32.65 | 0.08 | 73.94 | 0.19 |
|  |  | Low | 1523 | 0.93 | 100.29 | 0.38 | -1.54 | 0.00 | 73.21 | 0.17 |
|  |  | Clean | 1290 | 0.94 | 97.42 | 0.36 | -5.60 | -0.01 | 72.45 | 0.19 |



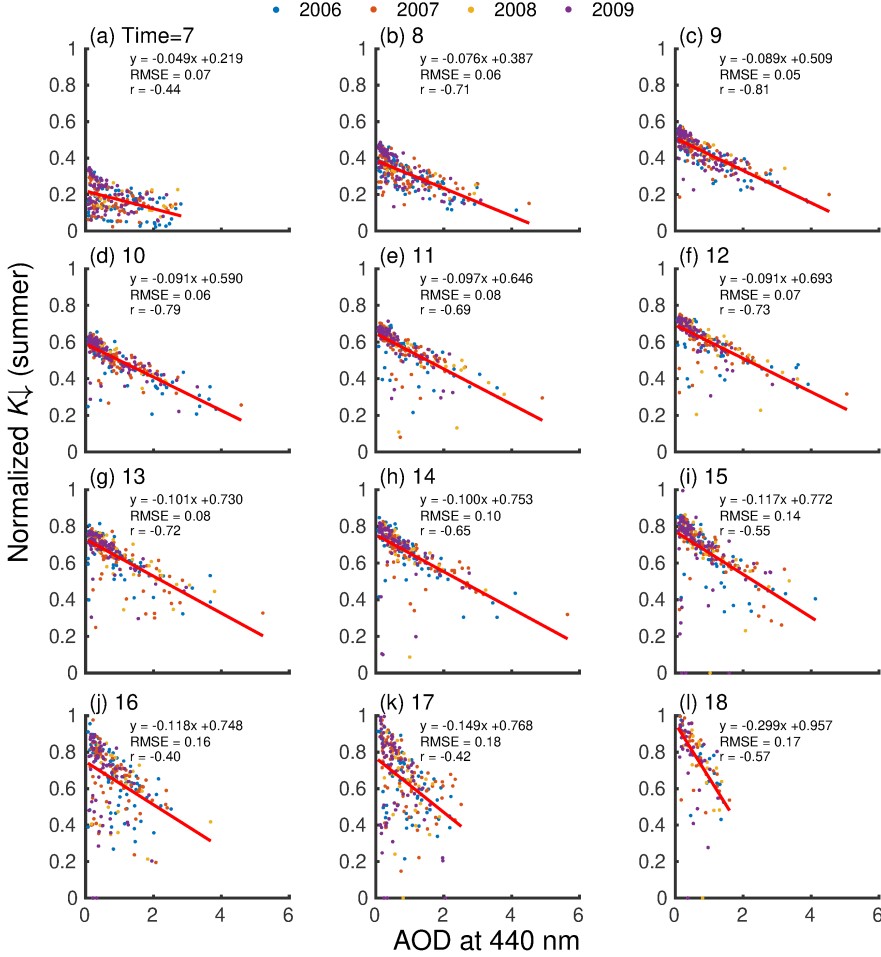

**Figure 1.** Observed hourly incoming solar radiation ($K\!\downarrow$) normalized with clear sky radiation ($I_{SC} \times \cos\theta_z$) against aerosol optical depth (AOD) observations (Holben et al., 1998) for different hours of day (a–l) for thermal summer months (Apr–Sept) at 325 m IAP meteorological tower (47 m level; Liu et al. (2012)). Winter months in Fig. A1. A linear regression (red line) is fitted after Lowess smoothing made for the scatter. For statistics see Sect. 2.1.

The WFDEI precipitation is higher than observed for days with <11 mm day$^{-1}$ of precipitation, but too low for higher (>11 mm day$^{-1}$) daily rainfall rates (Fig. 3). After the BCQM correction (Kokkonen et al., 2018b) the correspondence with observations is generally improved (Fig. 3) similarly to earlier results in Vancouver and London (Kokkonen et al., 2018b). However, $P_{WF}$ statistics (Table 3) during extremely polluted and polluted levels become slightly poorer ($r$: from 0.42 to 0.38 and 0.73 to 0.68; nRMSE: from 0.95 to 1.06 and 0.85 to 1.33, respectively) whereas mainly improving with low pollution and fairly clean pollution levels ($r$: from 0.49 to 0.55 and 0.66 to 0.67; nRMSE: from 0.95 to 1.00 and 0.80 to 0.77, respectively). It is expected that the correction affects mostly cleaner conditions since most of the larger daily totals of precipitation occur during low pollution and fairly clean air (Fig. 4).



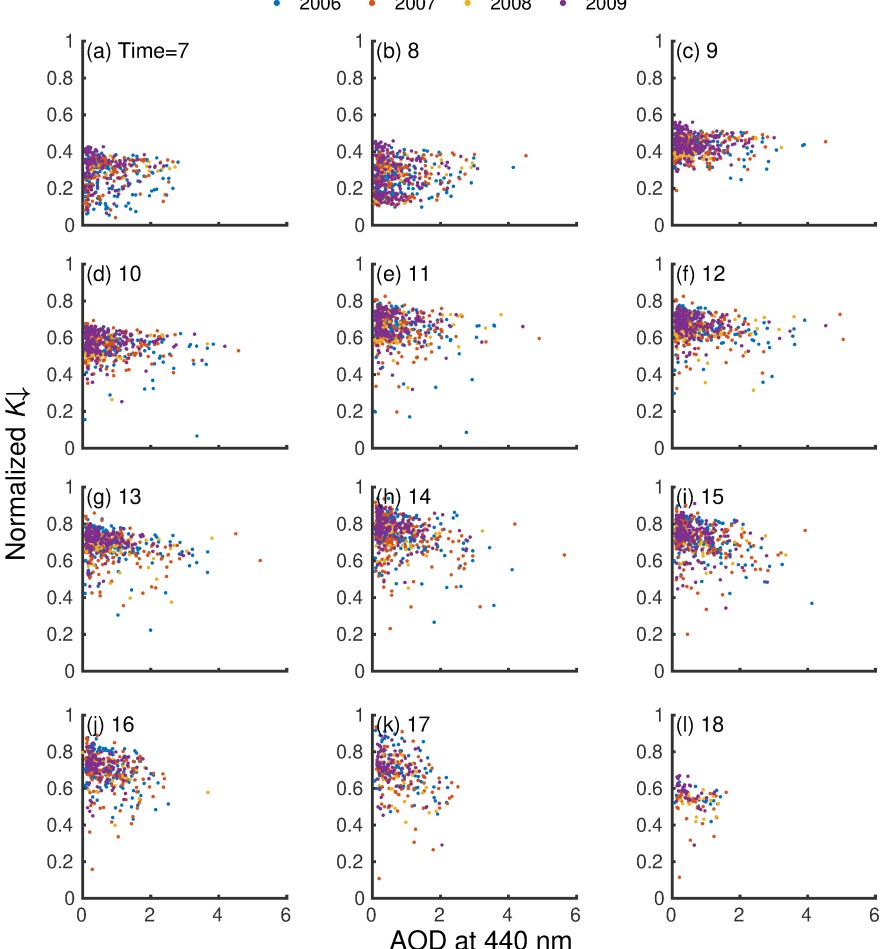

**Figure 2.** Uncorrected WFDEI incoming solar radiation ($K\downarrow$) downscaled from 3 h resolution to 1 h (Kokkonen et al., 2018b) normalized with clear sky radiation ($I_{SC} \times \cos\theta_z$) against AOD observations (Holben et al., 1998) for different hours of day (a–l) at 325 m IAP meteorological tower.

The corrected $K\downarrow_{WF}$ and other meteorological variables correspond well with observations in all air quality levels except for $P_{WF}$ which still has substantial biases even after the correction (Fig. 5, Table 3).

### 3.1.1 Meteorological conditions during haze

Haze events in Beijing typically occur with southerly wind, which brings warm and humid air masses from the south (e.g., Cai
5 et al., 2017; Chen and Wang, 2015; Wu et al., 2017). In addition, the wind speeds are typically slower ($< 2\ \mathrm{m\ s^{-1}}$) during haze events (Zheng et al., 2015, 2016). The highly polluted industrial areas are located south of Beijing, so southerly winds transport pollutants from these areas (Zhang et al., 2014; Zhao et al., 2013; Zheng et al., 2015). These meteorological conditions are also



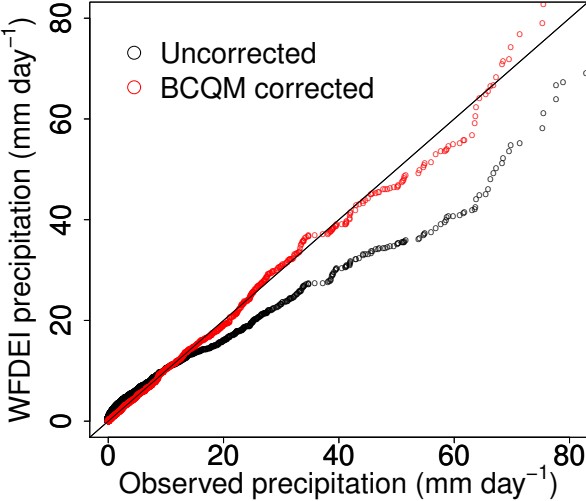

**Figure 3.** Quantile-quantile plot of uncorrected WFDEI precipitation and WFDEI precipitation bias corrected using quantile mapping (BCQM; Kokkonen et al. (2018b)) versus observed precipitation (1980–2012).

favorable to haze formation from local emissions and secondary nucleation (Kulmala et al., 2016; Yao et al., 2018). Due to wet deposition of aerosols the high precipitation rates commonly occur on less polluted days (Ouyang et al., 2015).

Meteorological conditions during haze events are well represented in the corrected WFDEI dataset. Figure 6 shows the average daily meteorological conditions of all the days when AOD >1 ($N$ = 568, day 0), five days before and five days after for different WFDEI meteorological variables in 2001–2013. When AOD increases in the extremely polluted conditions (from 0.99 in day -5 to 1.82 in day 0), $RH_{WF}$ and $T_{air,WF}$ also increase (from 55.1 % and 22.5°C in day -5 to 56.0 % and 24.0°C in day 0) and $U_{WF}$ and $P_{WF}$ decrease (from 1.8 m s$^{-1}$ and 2.5 mm day$^{-1}$ in day -5 to 1.8 m s$^{-1}$ and 1.0 mm day$^{-1}$ in day 0). The correct description of meteorological conditions during haze events makes the study of water balance during different pollution levels possible using the WFDEI data.

## 3.2 Evaluation of SUEWS model in polluted urban environment

SUEWS model performance is relatively independent of haze levels as its effects on local meteorological conditions are included in the model input variables $K\downarrow$, $T_{air}$ and RH. As the incoming longwave radiation ($L\downarrow$) emitted by the sky is calculated from $T_{air}$ and RH, which have positive correlation with level of pollution in Beijing (e.g., Cai et al., 2017; Chen and Wang, 2015; Wu et al., 2017), the positive correlation of $L\downarrow$ and air quality is reproduced by SUEWS (Figure 7b). Therefore, the model performance does not significantly decrease with increasing AOD (Fig. 5, Table 5), even though there are substantial differences in uncertainties of $P_{WF}$ between the different air quality conditions. Surface runoff, which is most sensitive to precipitation, is analysed using normalized values and therefore the uncertainties in $P_{WF}$ are not crucial to the conclusions.




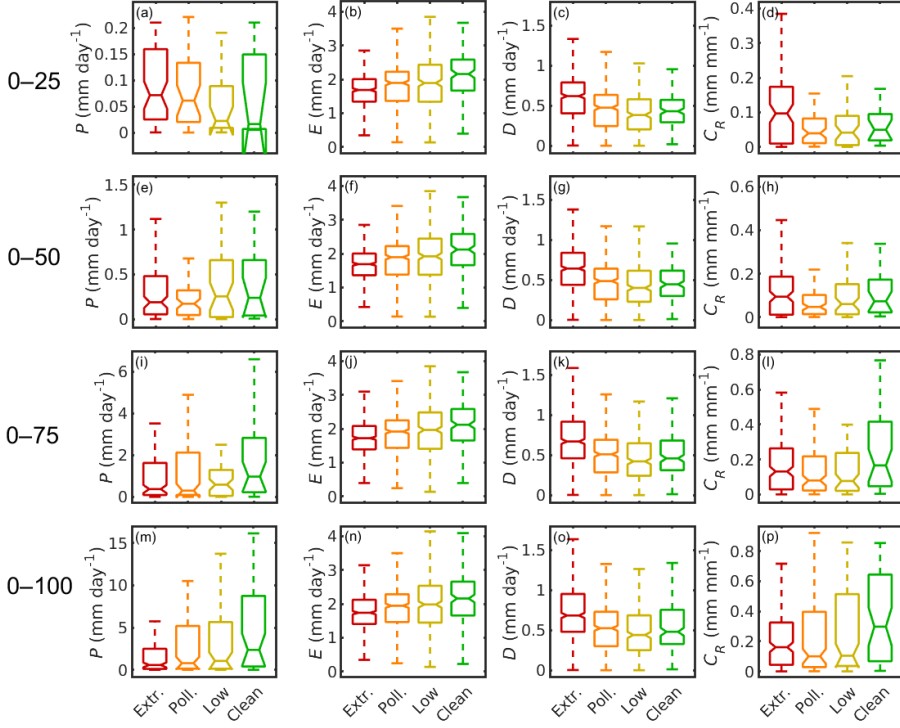

**Figure 4.** Box plots of daily precipitation ($P$), evapotranspiration ($E$), drainage ($D$) and runoff coefficient ($C_R$) stratified by different pollution levels (extremely polluted air, polluted air, low pollution, fairly clean air; see Sect. 2.1 for details) and daily precipitation percentiles (rows) from low (0–25) to all precipitation events (0–100) for 2001–2013. The notches indicate the 95 % confidence levels. Outliers are not shown. The amount of data used for each box is shown in Table 4. For statistics see Sect. 2.1.

After the above corrections are made to the WFDEI data, the model performance is improved (Table 5) and SUEWS simulates $Q_E$ well ($r>0.73$, nRMSE: 0.58 to 0.81 from clean to extremely polluted conditions) and the results during different air quality levels are generally comparable to each other (Fig. 5, Table 5). Also the modelled sensible heat flux ($Q_H$) is reasonably good ($r>0.74$, nRMSE: 0.83 to 1.33 from clean to extremely polluted conditions) with overestimating slightly the daytime

5   values. Similar overestimation has been observed with other urban local scale models used in Beijing (e.g., Liang et al., 2018) and relates likely to the overestimated anthropogenic heat flux ($Q_F$) or underestimated storage heat flux ($\Delta Q_S$) values that cannot be easily measured. Detailed hydrological analysis on the effect of haze can be made using SUEWS forced by WFDEI data in highly polluted Beijing as, the performance of the model is similar to the results in cleaner cities of Vancouver, Los Angeles, London and Swindon (e.g., Järvi et al., 2011; Kokkonen et al., 2018b; Ward et al., 2016).



**Table 4.** Number of days of data ($N$) in each Fig. 4 box plot stratified by pollution levels (extremely polluted air, polluted air, low pollution, fairly clean air; see Sect. 2.1 for details) and daily precipitation percentiles (columns) from low (0–25) to all precipitation events (0–100). For explanation of the statistical methods see Sect. 2.1.

|  |  | Precipitation percentiles | | | |
| --- | --- | --- | --- | --- | --- |
| Variable | Level of pollution | 0–25 | 0–50 | 0–75 | 0–100 |
| $P$ | Extreme | 65 | 110 | 152 | 175 |
|  | Polluted | 36 | 62 | 90 | 113 |
|  | Low | 21 | 42 | 55 | 70 |
|  | Clean | 8 | 19 | 33 | 46 |
| $E$ | Extreme | 458 | 503 | 545 | 568 |
|  | Polluted | 386 | 412 | 440 | 463 |
|  | Low | 280 | 301 | 314 | 329 |
|  | Clean | 148 | 159 | 173 | 186 |
| $D$ | Extreme | 457 | 502 | 544 | 567 |
|  | Polluted | 377 | 403 | 431 | 454 |
|  | Low | 270 | 291 | 304 | 319 |
|  | Clean | 145 | 156 | 170 | 183 |
| $C_R$ | Extreme | 94 | 132 | 174 | 197 |
|  | Polluted | 68 | 87 | 115 | 138 |
|  | Low | 41 | 60 | 73 | 88 |
|  | Clean | 17 | 27 | 40 | 52 |

## 3.3 The radiative effect of haze on surface water balance

Comparison of $K\downarrow$ on different pollutant levels shows that aerosols attenuate $K\downarrow_{WF}$ by 167 W m$^{-2}$ (medians of midday $K\downarrow_{WF}$ of fairly clean conditions and extremely polluted conditions; Fig. 7). This reduces surface energy availability and sensible heat flux (Kajino et al., 2017). In addition, $K\downarrow$ absorbed by the heavily polluted layer changes the vertical temperature profile leading to an increased stability, which reduces turbulence and mixing and therefore also the boundary layer height (Petäjä et al., 2016). With less energy available at the surface, evaporation decreases by 0.42 mm day$^{-1}$ (daily median of fairly clean compared to extremely polluted conditions, Fig. 4). Thus with the same precipitation rate more water would be stored at the surface and soil and directed to surface runoff especially during smaller precipitation intensities on more polluted levels (Fig. A2). The drainage is decreased by 0.19 mm day$^{-1}$ and the runoff coefficient ($C_R = R/[P+I]$) is increased by 0.047 based on the same comparison of daily medians for the 0–25 percentiles daily precipitation (Fig. 4). This is because for the most polluted days with lowest precipitation (0–25 percentiles) $P$ is slightly larger (0.07 mm day$^{-1}$) and $E$ is the lowest (1.69 mm day$^{-1}$)



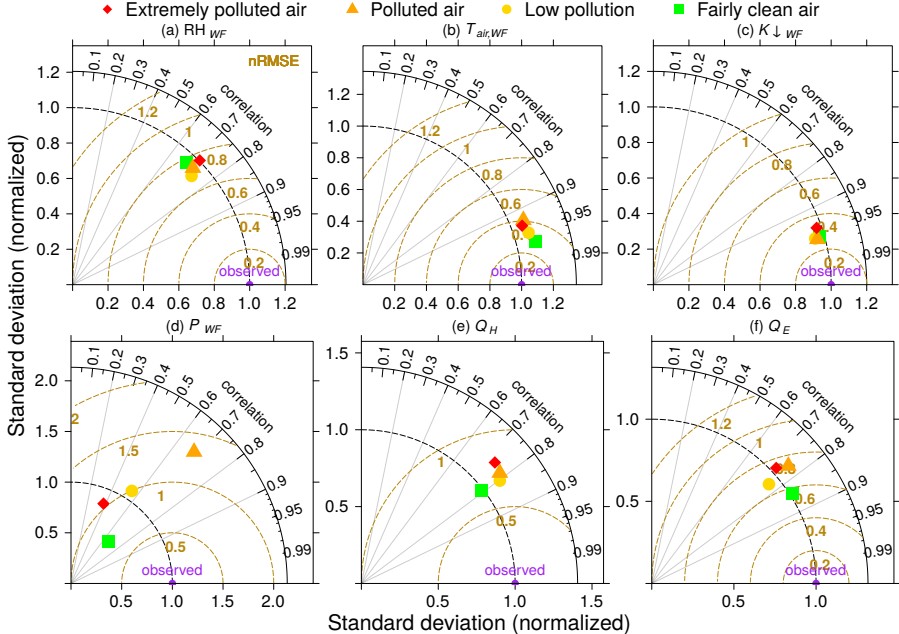

**Figure 5.** Taylor diagram (Taylor, 2001) for hourly (a) relative humidity ($RH_{WF}$), (b) air temperature ($T_{air,WF}$), (c) incoming solar radiation ($K{\downarrow}_{WF}$) and daily (d) precipitation ($P_{WF}$) with corrected WFDEI data assessed with IAP observations stratified by air quality (Sect. 2.1), and hourly modelled (e) sensible heat flux ($Q_H$) and (f) latent heat flux ($Q_E$) against eddy covariance IAP observations from 47 m height for 2006–2009. The radial axis is normalized standard deviation, angular axis is correlation coefficient and brown dashed lines indicate normalized root-mean square error (nRMSE). See Sect. 2.1 for statistics explanation.

resulting in $C_R$ being largest (median 0.097) whereas the cleaner conditions are substantially lower (0.039–0.049) (Fig. 4). As higher daily precipitation percentiles are included, the higher amount of $P$ during the fairly clean conditions starts to dominate. Even though the median $C_R$ during extremely polluted conditions is higher than during other polluted levels (polluted and low pollution conditions) in all of the precipitation classes, $C_R$ during fairly clean air starts to be equal with the extreme haze conditions during days with precipitation of 0–75 percentiles ($C_R$: 0.13, 0.08, 0.08, 0.17 for extremely polluted, polluted, low pollution and fairly clean air, respectively) and exceeds extreme haze conditions when all the percentiles of precipitation are included (0.16, 0.10, 0.10, 0.30 from extremely polluted to fairly clean conditions).

Beijing urban top soil is heavily polluted in gardens, roadsides and residential areas in Beijing (e.g., Chen et al., 2005; Xia et al., 2011). Two-thirds of Beijing's water supply is from groundwater which is often contaminated by surface pollution sources (Sun et al., 2014). Decreased evaporation with poorer air quality increases surface drainage (0.70, 0.54, 0.44, 0.50 mm day$^{-1}$ from extremely polluted to fairly clean conditions) (Fig. 4) potentially causing more infiltration to groundwater from surfaces on occasions where higher atmospheric deposition has occurred. Hence, potentially making water quality poorer.

The increase in surface runoff during high haze conditions is quite small (Fig. 4) and may not contribute significantly to urban flooding. However, the poorest surface runoff water quality is associated with first flush of runoff (Deletic and Maksimovic,





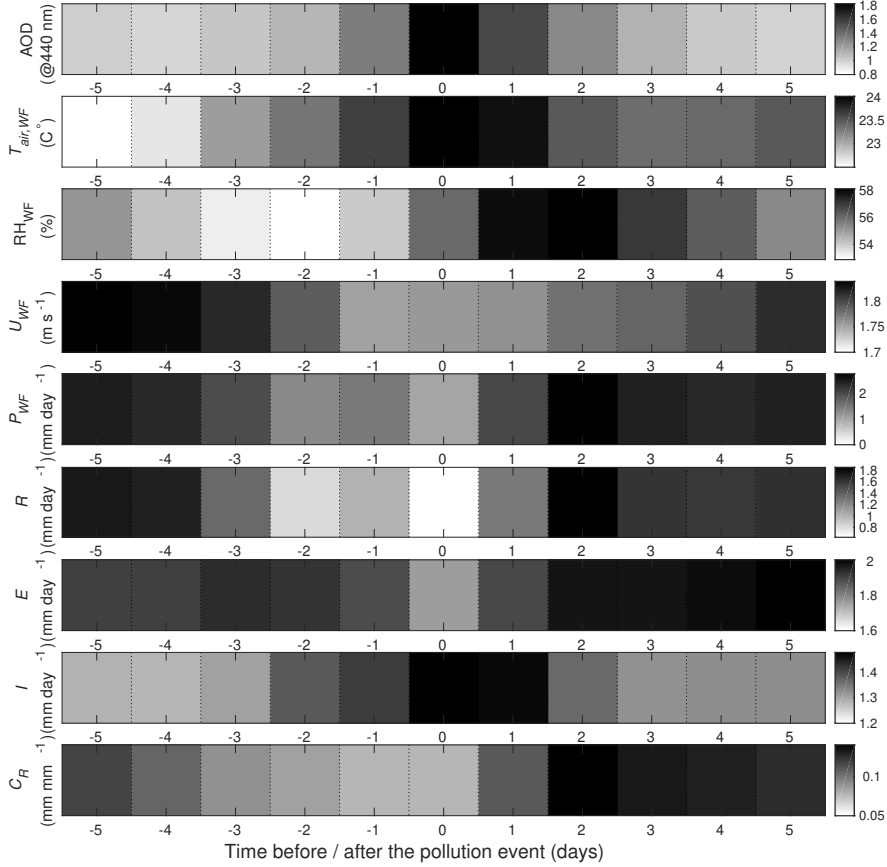

**Figure 6.** Average meteorological conditions of the pollution event days (AOD > 1, day 0), five days before (days -5 to -1) and five days after (days 1 to 5) in 2001–2013. Daily averages ($N = 568$) of aerosol optical depth (AOD), air temperature ($T_{air,WF}$), relative humidity (RH$_{WF}$), wind speed ($U_{WF}$) and daily cumulative values of observed precipitation ($P_{WF}$) and modelled surface runoff ($R$), evapotranspiration ($E$), irrigation ($I$) and runoff coefficient ($C_R$).

1998; Gupta and Saul, 1996; Yufen et al., 2008), thus the days with increased runoff may have poorer water quality. The irrigation of urban green areas might have also similar effect by flushing the pollutants regularly from the surfaces. The flush of pollutants from contaminated surfaces to urban water bodies as surface runoff from vegetated and impervious surfaces in Beijing has been shown to include significantly more pollutants than rain water (Yufen et al., 2008). Therefore, the increase in runoff and drainage due to radiative effect of air pollution will increase the pollutant loads in already deteriorated urban surface waters (Sun et al., 2014) and groundwater.



**Table 5.** Comparison of hourly model results of $Q_H$ and $Q_E$ for 2006–2009 stratified by pollution levels (extremely polluted air (AOD>1), polluted air (0.438–1), low pollution (0.203–0.438), fairly clean air (<0.203) (see Sect. 2.1 for details). Superscript $uc$ indicates uncorrected variables. For explanation of the statistical methods see Sect. 2.1.

| | Variable | Level of pollution | $N$ | $r$ | RMSE | nRMSE | MBE | nMBE | MAE | nMAE |
|---|---|---|---|---|---|---|---|---|---|---|
| Model results | $Q_H$ | Extreme | 848 | 0.74 | 62.27 | 1.33 | 49.76 | 1.03 | 52.58 | 1.09 |
| | | Polluted | 990 | 0.78 | 69.01 | 1.23 | 55.80 | 1.06 | 59.14 | 1.13 |
| | | Low | 896 | 0.80 | 73.87 | 1.05 | 56.71 | 0.77 | 62.09 | 0.84 |
| | | Clean | 837 | 0.79 | 57.80 | 0.83 | 36.32 | 0.48 | 47.34 | 0.62 |
| | $Q_H^{uc}$ | Extreme | 848 | 0.69 | 94.85 | 2.02 | 82.37 | 1.70 | 84.75 | 1.75 |
| | | Polluted | 990 | 0.72 | 84.36 | 1.51 | 69.52 | 1.32 | 74.31 | 1.42 |
| | | Low | 896 | 0.75 | 83.29 | 1.19 | 65.48 | 0.88 | 71.56 | 0.97 |
| | | Clean | 837 | 0.74 | 63.88 | 0.91 | 41.17 | 0.54 | 52.67 | 0.69 |
| | $Q_E$ | Extreme | 850 | 0.73 | 38.10 | 0.81 | 15.36 | 0.38 | 27.27 | 0.67 |
| | | Polluted | 995 | 0.76 | 41.65 | 0.76 | 11.07 | 0.23 | 27.66 | 0.58 |
| | | Low | 915 | 0.76 | 36.80 | 0.67 | -3.30 | -0.06 | 24.39 | 0.46 |
| | | Clean | 883 | 0.84 | 33.97 | 0.58 | -6.47 | -0.10 | 23.62 | 0.38 |
| | $Q_E^{uc}$ | Extreme | 850 | 0.68 | 43.56 | 0.93 | 18.86 | 0.46 | 31.66 | 0.78 |
| | | Polluted | 995 | 0.71 | 44.74 | 0.82 | 10.20 | 0.21 | 29.47 | 0.62 |
| | | Low | 915 | 0.74 | 38.50 | 0.70 | -7.32 | -0.14 | 26.42 | 0.49 |
| | | Clean | 883 | 0.83 | 35.43 | 0.60 | -9.90 | -0.16 | 25.53 | 0.41 |

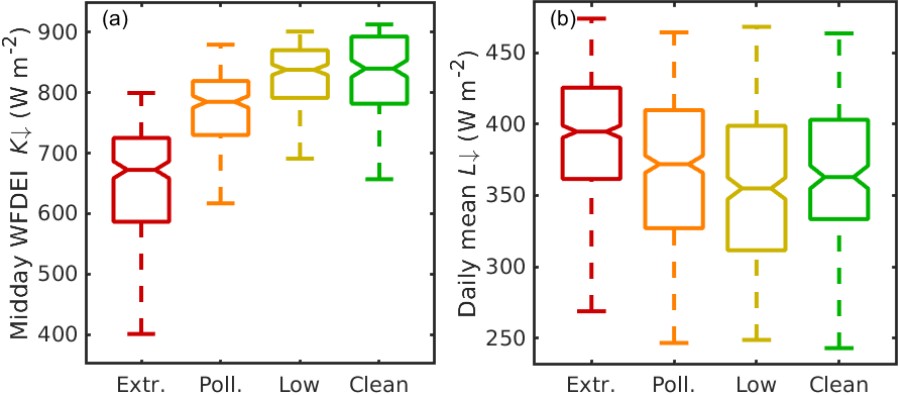

**Figure 7.** Box plots of (a) midday WFDEI incoming solar radiation ($K{\downarrow}_{WF}$) and (b) modelled daily mean incoming longwave radiation ($L{\downarrow}$) stratified by different pollution levels (extremely polluted air, polluted air, low pollution, fairly clean air) for 2001–2013. See Sect. 2.1 for details.





## 4 Conclusions

In this study the radiative effect of haze on local scale hydrological cycle is examined for the period 2001–2013. The hydrological modelling is conducted using Surface Urban Energy and Water Balance Scheme (SUEWS) forced with WATCH WFDEI reanalysis data. The representativeness of WFDEI reanalysis data in highly polluted urban environment (Beijing) is assessed

with meteorological observations from 325 m IAP tower from 47 m level. In addition, the SUEWS performance is evaluated against eddy covariance observations of latent and sensible heat fluxes from the same height of IAP tower. The results are stratified by air quality based on observations of aerosol optical depth.

The effects of pollution are well accounted for in the original WFDEI meteorological variables except for incoming solar radiation and precipitation. After the correction, daily precipitation totals are generally improved, but there are still substantial

differences in the performance between the different air quality levels. After correcting $K\!\downarrow_{WF}$ with new developed haze correction, it compares well to observations across pollution levels ($r$ >0.94, nRMSE <0.33). Evaluations of SUEWS using eddy covariance observations of evaporation in Beijing concludes the model performance is good ($r$: >0.68 and >0.73; nRMSE: <0.93 and <0.81 using uncorrected and corrected WFDEI forcing data, respectively). Similarly SUEWS performance of sensible heat flux is rather good ($r$: >0.69 and >0.74; nRMSE: <2.02 and <1.33 using uncorrected and corrected WFDEI forcing

data, respectively). Therefore the local urban water balance can be modelled despite substantial biases in WFDEI precipitation data.

Detailed analyses of water balance terms finds that attenuated $K\!\downarrow$ from atmospheric pollution decreases the daily median evapotranspiration from 2.16 mm day$^{-1}$ during fairly clean conditions to 1.74 mm day$^{-1}$ during extremely polluted conditions. This leads to an increased runoff coefficient (from 0.049 to 0.097 during fairly clean and extremely polluted conditions,

respectively) especially during smaller precipitation totals (days with precipitation totals of 25th percentile). When all precipitation events are included the higher precipitation levels during fairly clean conditions induce highest runoff coefficients (0.30), even though the runoff coefficient during the extremely polluted conditions (0.16) is higher than during other air quality levels (0.10 in both polluted and low pollution conditions). Also soil infiltration is increased due to decreased evapotranspiration: drainage from 0.48 mm day$^{-1}$ during fairly clean conditions to 0.68 mm day$^{-1}$ during extremely polluted conditions.

This study is the first to examine the radiative effects of haze on local scale urban hydrological cycle. The increased surface runoff and soil infiltration are expected to lead to increased pollutant loads washed from polluted surfaces and top layers of soils into urban surface waters and groundwater which are already poor in the Beijing region. The evaluation of WFDEI reanalysis data gives first results of the representativeness of an reanalysis dataset in a highly polluted urban area. Also other reanalysis datasets should be carefully evaluated and make the needed corrections prior to use at polluted urban areas.

*Code and data availability.* For SUEWS manual and software, visit: http://suews-docs.readthedocs.io. WATCH WFDEI data can be acquired from ftp://rfdata:forceDATA@ftp.iiasa.ac.at and AOD data from https://aeronet.gsfc.nasa.gov/.



## Appendix A

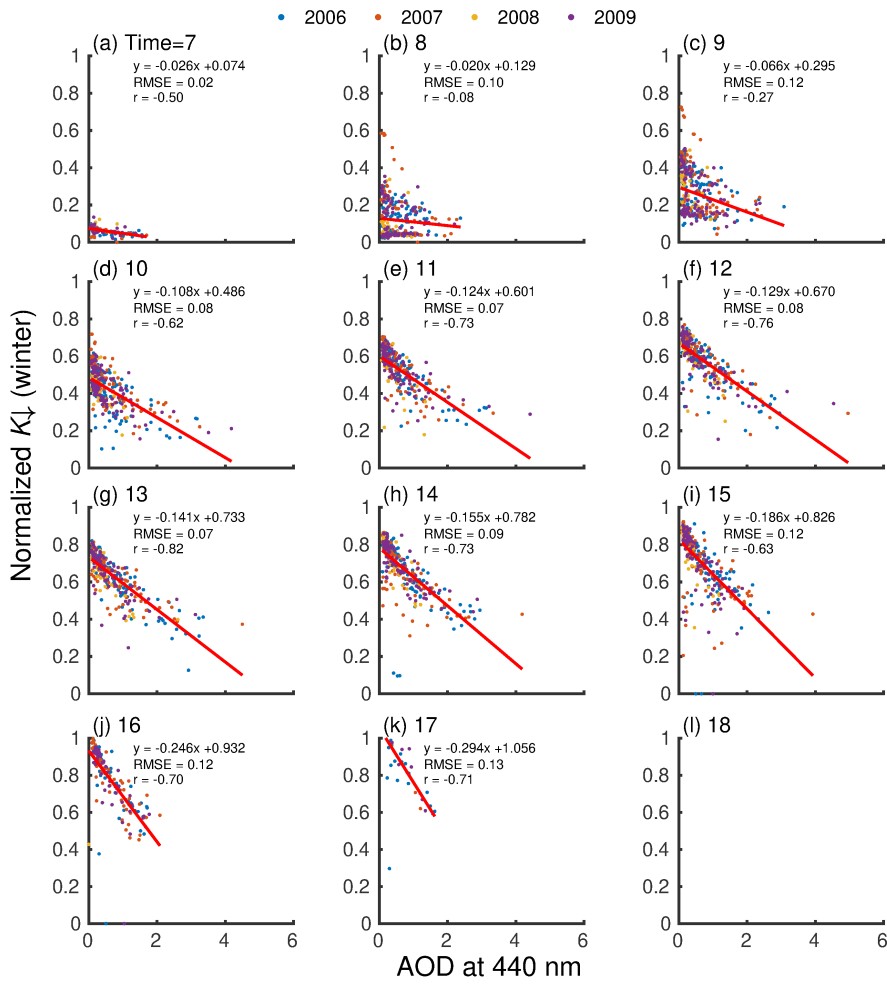

**Figure A1.** As in Fig. 1, but for thermal winter months (Oct–Mar).





**Table A1.** Notations used in Tables 1 and A2. Details and sources of the values in Järvi et al. (2011, 2014).

| Variable | Description | Variable | Description |
|---|---|---|---|
| $\alpha_i$ | Effective surface albedo | DaysSinceRain | Days since rain before the simulation period (from WATCH data of previous year) |
| $\alpha_s$ | Effective snow albedo | | |
| $\alpha_s^{min}$ | Minimum snow albedo | $G_{1-6}$ | Parameters related to surface conductance |
| $\alpha_s^{max}$ | Maximum snow albedo | GDD | Growing degree days |
| $\epsilon_i$ | Effective surface emissivity | $g_{i,\max}$ | Maximum conductance (m s$^{-1}$) |
| $\epsilon_s$ | Effective surface emissivity | $I_w$ | Additional water to water surface type (mm) |
| $\rho_e$ | Threshold value in the calculation of retention capacity (kg m$^{-3}$) | $K\downarrow_m$ | Maximum incoming solar radiation used in $g_s$ calculation |
| $\rho_s^{min}$ | Minimum snow density (kg m$^{-3}$) | $\text{LAI}_{\max}$ | Maximum LAI of surface type $i$ (m$^2$ m$^{-2}$) |
| $\rho_s^{max}$ | Maximum snow density (kg m$^{-3}$) | $\text{LAI}_{\min}$ | Minimum LAI of surface type $i$ (m$^2$ m$^{-2}$) |
| $\tau_a$ | Cold snow time constant for snow albedo aging | $r_s^{max}$ | Maximum surface resistance (s m$^{-1}$) |
| $\tau_f$ | warm snow time constant for snow albedo aging | $\text{res}_{\text{cap}}$ | Surface water capacity in LUMPS (mm) |
| $a_{0,\{\text{wd,we}\}}$ | Parameter defining the base $Q_F$ per capita (W m$^{-2}$ (capita$^{-1}$ ha$^{-1}$)$^{-1}$) | $\text{res}_{\text{drain}}$ | Drainage rate of water bucket in LUMPS (mm h$^{-1}$) |
| $a_{1,\{\text{wd,we}\}}$ | Parameter defining the base CDD per capita (W m$^{-2}$ K$^{-1}$ (capita$^{-1}$ ha$^{-1}$)$^{-1}$) | $R_C$ | Limit when surface is totally covered with water in LUMPS (mm) |
| $a_{2,\{\text{wd,we}\}}$ | Parameter defining the base HDD per capita (W m$^{-2}$ K$^{-1}$ (capita$^{-1}$ ha$^{-1}$)$^{-1}$) | $S_{1,2}$ | Parameters related to surface conductance |
| $a_{1,2,3}$ | Constants in the calculation of the snow heat storage | $S_i$ | State of the snow-free surface (mm) |
| | | $S_{\text{soil},i}$ | Soil state (mm) |
| | | $S_{\text{pipe}}$ | Maximum depth capacity of pipes (mm) |
| $a_f$ | Temperature freezing factor (mm $^\circ$C$^{-1}$ h$^{-1}$) | SDD | Senescence degree days |
| $a_r$ | Radiation melt factor (mm W$^{-1}$ h$^{-1}$) | $\text{SWE}_{\max,i}$ | Snow water equivalent when surface type $i$ is fully covered with snow (mm) |
| $a_t$ | Temperature melt factor (mm $^\circ$C$^{-1}$ h$^{-1}$) | | |
| $b$ | Empirical coefficient in the calculation of drainage | $\text{SWE}_{\text{lim}}$ | Limit of the snow water equivalent for snow removal (mm) |
| $b_{0a,1a,2a}$ | Parameters for automatic irrigation (mm, mm K$^{-1}$, mm d$^{-1}$) | $T_{air}^{\text{initial}}$ | Initial air temperature ($^\circ$C) |
| | | $T_{\text{BaseGDD}}$ | Base temperature for leaf growth ($^\circ$C) |
| $b_{0m,1m,2m}$ | Parameters for automatic irrigation (mm, mm K$^{-1}$, mm d$^{-1}$) | $T_{\text{BaseSDD}}$ | Base temperature for senescence ($^\circ$C) |
| | | $T_{\text{BaseQF}}$ | Base temperature for $Q_F$ ($^\circ$C) |
| $C_i$ | Interception state of $i$th surface (mm) | $T_{\text{lim}}$ | Temperature limit for the liquid precipitation and snow ($^\circ$C) |
| $C_{soil,i}$ | Soil water storage (mm) | | |
| $C_{min}^R$ | Minimum retention capacity (mm) | $T_H, T_L$ | Parameters related to calculation of $g_s$ ($^\circ$C) |
| $C_{max}^R$ | Maximum retention capacity (mm) | $T_{\text{step}}$ | Time step for water balance calculation (s) |
| $D_{0,i}$ | Drainage rate (mm) | $z_{soil}$ | Depth of the soil layer (mm) |



**Table A2.** Model parameters used in SUEWS for different surfaces: buildings (bldgs), paved (pav), evergreen vegetation (everg), deciduous vegetation (dec), grass and water. Initial conditions assume there is no snow on the ground and leaf area index of each vegetation type is at their minimum value. See Table A1 for notation and Järvi et al. (2011, 2014) for data sources.

|  | Units | Bldgs | Pav | Everg | Dec | Grass | Water |
|---|---|---|---|---|---|---|---|
| $S_i$ | mm | 0.25 | 0.48 | 1.3 | 0.3–0.8 | 1.9 | 0.5 |
| $S_{\text{soil},i}$ | mm | 150 | 150 | 150 | 150 | 150 | – |
| $D_{0,i}$ | mm | 10 | 10 | 0.013 | 0.013 | 10 | – |
| $b$ | – | 3 | 3 | 1.71 | 1.71 | 0.013 | – |
| $C_i$ | mm | 0 | 0 | 0 | 0 | 0 | 0 |
| $C_{\text{soil},i}$ | mm | 30 | 70 | 130 | 130 | 130 | – |
| $\alpha_i$ | – | 0.15 | 0.12 | 0.1 | 0.16 | 0.19 | 0.1 |
| $\epsilon_i$ | – | 0.95 | 0.91 | 0.98 | 0.98 | 0.93 | 0.95 |
| $g_{i,\text{max}}$ | mm s$^{-1}$ | – | – | 7.4 | 11.7 | 40 | – |
| $\text{LAI}_{\text{max}}$ | m$^2$ m$^{-2}$ | – | – | 5.1 | 5.5 | 5.9 | – |
| $\text{LAI}_{\text{min}}$ | m$^2$ m$^{-2}$ | – | – | 4.0 | 1.0 | 1.6 | – |
| $\text{SWE}_{\text{max},i}$ | mm | 190 | 190 | 190 | 190 | 190 | – |
| $\text{SWE}_{\text{lim}}$ | mm | 40 | 100 | – | – | – | – |
| $z_{soil}$ | mm | 349 | 349 | 349 | 349 | 349 | – |

**Table A3.** Disaggregation for precipitation parameters (see details from Ward et al. (2017, 2018)).

| Resolution of input | 3 h | | |
|---|---|---|---|
| Disaggregation method | 102 | | |
| RainAmongN | 36 | | |
| MultRainAmongN | 15 | 24 | 36 |
| MultRainAmongNUpperI | 1.5 | 6.0 | 150.0 |

Rainfall is evenly distributed among RainAmongN subintervals in a rainy interval for different intensity bins. The number of subintervals over which to distribute rainfall in each interval is given in MultRainAmongN for three intensity bins. Upper limit for each intensity bin to apply MultRainAmongN is given in MultRainAmongNUpperI.





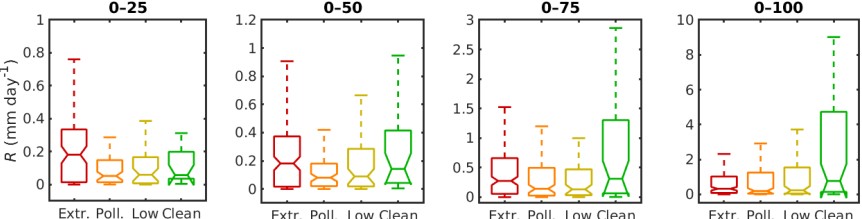

**Figure A2.** Daily cumulative runoff ($R$) stratified by different pollution levels (extremely polluted air, polluted air, low pollution, fairly clean air; see Sect. 2.1 for details) and daily precipitation percentiles from low (0–25) to all precipitation events (0–100) for 2001–2013. The notches indicate 95 % confidence levels. Outliers are not shown. For statistics see Sect. 2.1. See also Fig. 4.



*Author contributions.* TVK and LJ conceived this study; TVK was responsible for the atmospheric and hydrological analyses; SM was responsible for the GIS analysis; HL was responsible for the meteorological measurements. All authors contributed in writing the manuscript.

*Competing interests.* The authors declare that they have no conflict of interest.

*Acknowledgements.* We thank Maa- ja vesitekniikan tuki ry for funding (grant number 36663). We thank also Newton Fund/Met office
5  CSSP-China (Grimmond). We also acknowledge the city of Beijing for providing the sites for measurements and Hong-Bin Chen and Philippe Goloub for providing the AOD measurements. The data used are listed in the references.





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
