# Peer review of "Simulation of the radiative effect of haze on urban hydrological cycle using reanalysis data in Beijing"

_Atmospheric Chemistry and Physics, 2018_

## Referee Comment (RC1) · Anonymous Referee #2 · 21 Feb 2019

General comments By evaluating the impact of haze on urban hydrological cycle and limitation of the current modelling multi-scale approach, the paper addresses a relevant scientific issue that will help the scientific communities and decision makers worldwide. This research gains even more in importance in highly dense Asian megacities (China, India) that already suffer from aerosol pollution.  By talking modelling approach and atmospheric chemical and physical processes, the reviewer agrees that the topics entirely fit with the concerns of the Atmospheric Chemistry and Physics journal. In general the manuscript is well written but effort can be make to simplify the sentences (sometime confusing) and on the abstract/introduction to clarify the aims (as it seems there are several) and take-out messages of the research. It seems that the paper questions

the modelling approach and particularly the quality of the global reanalysis data used to simulate the local urban hydrological cycles during haze episodes, the SUEWS urban land surface model performance as well as the interactions between the aerosols and the urban hydrological. This should be clearly stated from the abstract until the result sections. The novelty (the focus on the local scale), challenges, and operational urban water management implication raised in the paper should also be better justified in the introduction and generic terms should be avoid to go directly to the fact (aerosols instead of pollution) and determinant physical interactions treated by the papers. The introduction can be elaborated so as to immediately focus the reader on the nature of the pollution the authors are dealing with (aerosols and wet haze?) instead to use generic terms. It will help to strengthen the message of the introduction. Following are specific and some technical comments/corrections that will hopefully be helpful to the authors.

Specific comments Page 3, l.8. Is it possible to explain the specificity of the Murto (2017)'s methods. Murto (2017) does not detail enough the method behind the land cover model construction and how the various vegetation compositions are retrieved from the aerial photographs. • What is the benefit of using the Murto's method and two source of spatial information? • What is the resolution of the World imagery? Why a semi supervised classification was not able to distinguished evergreen from deciduous trees based on irradiance and trees from shrubs based on a structural geometry algorithm? • What is the quality of the OSM data in the region? Page 4 Can you confirm that the model has been run for a time period of 3 years and a 5 min time step over a 1km2 simulation domain? Page 5 l.28. What was the nMBE before the correction? Page 8, l.1. "SUEWS model performance is relatively independent of haze level (. . .) in the model input variables". • Is the precipitation not also an input variable affected by the haze levels? How the bias in the precipitation can impact the quality of the simulation with respect to the precipitation rates (p.8 l5.)? • What about the influence of haze level on the longwave radiations, surface temperature and resulting QH and atmosphere stability? • Should the model performance only

be evaluated with the evaporation? Evapotranspiration is the common term in the energy and precipitation budget but as the incoming energy is partitioned also amongst other terms (sensible, storage), does it worth it to also consider these variable in the evaluation of the simulations. A fortiori, aerosols have been proven to increase the contributions of the scattered radiation versus direct radiation in the solar energy budget, while they potentially absorb and emit longwave radiation resulting in heat retention in the atmosphere.

Table 2. What are the uncertainties associated with the temperature, humidity and wind speed sensors developed by the institute of Atmospheric Physics? As being nonstandard instruments, is it possible to have a description of these and know if they have been already tested against standard sensors? Page 10 l.11. Is surface runoff not diminished for small precipitation intensities compared to high precipitation intensity episodes? The infiltration capacity of the soil horizon is usually reduced during high precipitation intensity episodes due to the destruction of soil aggregates -> less porosity, and usually deeper wetting front in the soil resulting in higher surface resistance. Please clarify.

Technical corrections Abstract l. 5-6 please, rephrase. Additionally, it was not clearly stated before that the evaluation of the SUEWS performance is also part of the aim of the paper. It can be good either to neglect this aspect in the abstract or if crucial add this additional and somehow "hidden" aim in the abstract. l. 10-12 "induce" instead "induces" considering the plural "rates". Also the message of the sentence is a bit confused to figure in the abstract. Please simplify your message. l.11-12 this is a justification of the research, isn't it? It should maybe be placed before the general outcomes. Introduction Page 1, l. 17. Is "northeast China one of the most populated areas" a consequence of environmental problems?. Please rephrase. Page 2, l. 1-2. Please rephrase. The sentence is difficult to read in my opinion. Page 2. l 4-5. It can worth it to elaborate more the interactions between the aerosols, the solar radiation, and the boundary layer height and stability. How these elements interact? Further

[Figure]

Page 4 l.10. "as the main focus. . .balance". This information can be removed and is not so necessary here. Page 4 l.24 Please indicate where are the mentioned box-plots, RMSEs, etc. in the paper? Page 5 l-9-11. This is more appropriated in the introduction. Page 6 Table 3. Please modify the title to better explain what is presented in this table and which variables are there inter-compared. Page 11 l.11 –Page 12 l.11. This is an interesting discussion although not a result.

---

## Referee Comment (RC2) · Anonymous Referee #1 · 18 Mar 2019

The authors evaluated a reanalysis data and a hydrological model with observational data and examined the effects of haze impact on the surface hydrology. The examination of how haze impacts on local scale urban hydrological cycle is particularly interesting. But I have some concerns about the connection between two parts. There also needs some clarifications about the approach and the hydrological model. My specific comments are as below. I recommend a major revision before the paper can be accepted for a publication in ACP.

1. Part 1, Introduction

The authors did not describe the ways how urban pollution can impact on the hydrological cycle. On Page 2, 2nd paragraph, the authors mentioned the air temperature change by urban pollution, but the focus of the paper is how urban pollution impacts on hydrological cycle. It is necessary to provide some background about how precipitation or water on ground are connected with pollution in atmosphere.

In the introduction, it needs to be clear that the pollution effects on precipitation are not considered, which would add the uncertainty to the study. It seems to me that the paper only considers the effects induced by changed surface temperature. Pollution can change temperature profile in atmosphere and serve as cloud condensation nuclei to affect precipitation rate and then surface hydrology. But all of these are not considered and the paper needs to be clear about it.

Last paragraph of this section: need to be clear about which method is used for each objective.

2. Section 2

It is not clear that what are the physical parameters for model input and outputs. Although Table shows some input variables, those are just symbols and their physical meaning is not provided. Most importantly, there is no information about how aerosols/pollution would impact temperature, runoff, and soil infiltration in the model? This is very important to understand what mechanisms are included for pollution impact runoff and soil infiltration. For the hydrological model, is precipitation rate changed by any factors? Is it just an input, not an output?

3. Section 3

It is not clear to me how section 3.1 and 3.2 are relevant or connected. Or are they just separated results without much connection? Why not run the model validated in Section 3.1 to look at the effect of haze on surface water in Section 3.2? In this way, the connection between the two sections is clear.

Some symbols are used without being defined in the text, such as P, K,. . .

**4. Section 4**

Please use physical terms, not symbols in the conclusion text.

---

## Author Comment (AC1) · 4 May 2019

We thank the reviewers for their comments, these have improved the paper. Our detailed responses are given in blue below.

Page number and line numbers indicated in the responses are the new ones unless otherwise indicated

**Referee #1**

General comments

By evaluating the impact of haze on urban hydrological cycle and limitation of the current modelling multi-scale approach, the paper addresses a relevant scientific issue that will help the scientific communities and decision makers worldwide. This research gains even more in importance in highly dense Asian megacities (China, India) that already suffer from aerosol pollution. By talking modelling approach and atmospheric chemical and physical processes, the reviewer agrees that the topics entirely fit with the concerns of the Atmospheric Chemistry and Physics journal. In general the manuscript is well written but effort can be make to simplify the sentences (some-time confusing) and on the abstract/introduction to clarify the aims (as it seems there are several) and take-out messages of the research. It seems that the paper questions the modelling approach and particularly the quality of the global reanalysis data used to simulate the local urban hydrological cycles during haze episodes, the SUEWS urban land surface model performance as well as the interactions between the aerosols and the urban hydrological. This should be clearly stated from the abstract until the result sections. The novelty (the focus on the local scale), challenges, and operational urban water management implication raised in the paper should also be better justified in the introduction and generic terms should be avoid to go directly to the fact (aerosols instead of pollution) and determinant physical interactions treated by the papers. The introduction can be elaborated so as to immediately focus the reader on the nature of the pollution the authors are dealing with (aerosols and wet haze?) instead to use generic terms. It will help to strengthen the message of the introduction. Following are specific and some technical comments/corrections that will hopefully be helpful to the authors.

*Response*
*Abstract changes*
The justification of the study is moved from the end of the abstract to the first paragraph and rephrased (page 1, l. 2–4): "Changes in hydrological cycle modify surface runoff and flooding. Furthermore, as runoff commonly transports pollutants to soil and water, any changes impact urban soil and aquatic environments."

Also the aims of the study (evaluation of the model and reanalysis data and the simulation of hydrological cycle) are rephrased to clarify it.

*Introduction changes*
The introduction text is rephrased to raise the novelty of local scale and the potential deterioration of urban water bodies (page 2, l. 14–17): "The higher surface runoff rates due to the modified water balance may increase pollutant loads in urban water bodies, by flushing pollutants from contaminated surfaces. However, the linkage between increased aerosol concentrations and urban hydrological cycle has not yet been studied in local scale despite its potential contribution to deterioration of urban aquatic environment."

The atmospheric (or air) pollution is rephrased in introduction to increased atmospheric aerosol concentration (page 1, l. 18, page 2, l. 7 and 16).

Description of physical interactions of pollutant concentrations, solar radiation, boundary layer height and its effects on water balance is added to the introduction (page 2, l. 10–14).

A short description of the importance of the accuracy of the reanalysis data is added (page 2, l. 21–24).

Specific comments

1. Page 3, l.8. Is it possible to explain the specificity of the Murto(2017)'s methods. Murto (2017) does not detail enough the method behind the landcover model construction and how the various vegetation compositions are retrieved from the aerial photographs.
The methods are widely used methods with ArcGIS-software using OpenStreetMap (OSM) and World Imagery, as described in the text (page 4, l. 2–7), and not specific to this study. With the available imagery the vegetation composition and tree height is not possible to obtain with GIS-methods. Therefore, it is estimated from the common tree species in Haidian district as described in page 4, l. 7–10.

2. What is the benefit of using the Murto's method and two source of spatial information?
There are substantial gaps in the OSM data in the region, especially at the southern part of the study area. Therefore two sources were used to evaluate the existing data at the areas where the two sources could be compared. This information is added to the text (page 4, l. 5–7).

3. What is the resolution of the World imagery?
The spatial resolution is 1m. This has been added to the text (page 4, l. 4).

4. Why a semi supervised classification was not able to distinguished evergreen from deciduous trees based on irradiance and trees from shrubs based on a structural geometry algorithm?
It might have been possible to distinguish the evergreen and deciduous trees from irradiance data, but from the data used this was not possible. However, the method used, where the classification is made using the studied proportions of the trees in the district should give high enough accuracy.

The fraction of trees and shrubs are given to the model as one input variable where both are included (for evergreen and deciduous separately) so there is no need to identify trees from shrubs.

5. What is the quality of the OSM data in the region?
See, comment #2 above.

6. Page 4 Can you confirm that the model has been run for a time period of 3 years and a 5 min time step over a 1km2 simulation domain?
The evaluation of the WFDEI data and the SUEWS model has been made for a time period of 4 years (2006–2009), since it is the time when observations were available. This is stated at page 2, l. 25 and 28 and page 4, l. 23–26. For the analysis of the hydrological cycle, the model is run for the period of 14 years (2000–2013), where the first year is used as a spin-up period, leaving years 2001–2013 for the analysis (page 4, l. 29). The time step for the model runs is 5 min. The study area is a 1 km radius circle around the IAP tower (page 3, l. 17–18) (i.e. area= 3.14 km$^2$).

7. Page 5 l.28. What was the nMBE before the correction?
The nMBE before correction (-0.01, 0.00, 0.08, 0.28 from clean to extremely polluted conditions) were added to the text (page 6, l. 13).

8. Page 8, l.1. "SUEWS model performance is relatively independent of haze level (...) in the model input variables". Is the precipitation not also an input variable affected by the haze levels?
Precipitation is added to the variables affected by the haze (page 9, l. 3).

9. How the bias in the precipitation can impact the quality of the simulation with respect to the precipitation rates (p.8 l5.)?
The surface runoff is the most sensitive to precipitation, but it is analysed using normalized values and therefore the uncertainties in precipitation are not crucial to the conclusions (page 9, l. 7–8). Also Fig. 5 and Table 5 show that the model performance is not substantially decreasing with increasing air pollution.

10. What about the influence of haze level on the longwave radiations, surface temperature and resulting QH and atmosphere stability?
The effect of haze on longwave radiation is shown in Fig. 7 and it is discussed in the text (page 9, l. 3–5 and page 2, l. 9).

The evaluation against the sensible heat flux which is also related to surface temperature is included in Fig. 5 and Table 5 showing the statistics. It is discussed in the text (page 10, l. 1–5 and page 14, l. 4–6).

The further analysis of QH and the stability of the atmosphere is out of the scope of this paper, as we focus on the hydrological cycle and surface processes. However, a short description of the effect of aerosols on atmospheric stability is included in page 2, l. 10–11 and page 10, l. 11–12.

11. Should the model performance only be evaluated with the evaporation? Evapotranspiration is the common term in the energy and precipitation budget but as the incoming energy is partitioned also amongst other terms (sensible, storage), does it worth it to also consider these variable in the evaluation of the simulations.
The model evaluation against sensible heat flux is included (see above comment #10). Unfortunately, the evaluation against storage heat flux is not possible due to difficulty of measuring this flux.

12. A fortiori, aerosols have been proven to increase the contributions of the scattered radiation versus direct radiation in the solar energy budget,while they potentially absorb and emit longwave radiation resulting in heat retention in the atmosphere.
Short description of this has been added (see above comment #10)

13. Table 2. What are the uncertainties associated with the temperature, humidity and wind speed sensors developed by the institute of Atmospheric Physics? As being nonstandard instruments, is it possible to have a description of these and know if they have been already tested against standard sensors?
We do not have detailed description of these instruments, but these have been widely used in previous internationally peer reviewed papers (e.g., Liu et al., (2012) Atmos. Chem. Phys. 12, 7881–7892; Song and Wang, (2012) Atmos. Res. 106, 139–149; Shi et al., (2018) Atmospheric and Oceanic Science Letters 12, 41–49; Al-Jiboori and Hu, (2005) Adv. Atmos. Sci. 22, 595–605)

14. Page 10 l.11. Is surface runoff not diminished for small precipitation intensities compared to high precipitation intensity episodes? The infiltration capacity of the soil horizon is usually reduced during high precipitation intensity episodes due to the destruction of soil aggregates -> less porosity, and usually deeper wetting front in the soil resulting in higher surface resistance. Please clarify.

Yes the surface runoff is smaller with small daily precipitation totals (as also seen in Fig. 4 and A2) for the above-mentioned reasons. However, surface runoff does not totally vanished as there are a lot of impervious surfaces (buildings and paved surfaces) that still generate surface runoff.

Technical corrections

15. Abstract l. 5-6 please, rephrase.
"We show how the reanalysis radiation data do not include the attenuating effect of haze and develop a haze correction for the incoming solar radiation. With this haze correction the SUEWS model simulates the eddy covariance measured latent heat flux well."
Rephrased as (abstract, l. 8–9):
"We show that the reanalysis data do not include the attenuating effect of haze on incoming solar radiation and develop a correction method. Using these corrected data, SUEWS simulates measured eddy covariance heat fluxes well."

16. Additionally, it was not clearly stated before that the evaluation of the SUEWS performance is also part of the aim of the paper. It can be good either to neglect this aspect in the abstract or if crucial add this additional and somehow "hidden" aim in the abstract.
In abstract it is rephrased (l. 6–7): "The secondary aims are to examine the usability of global reanalysis dataset in highly polluted environment and the SUEWS model performance."
In addition the introduction is rephrased as (page 2, l. 25–32):
"The aims of this study are (1) to evaluate the 2006 to 2009 WATCH Forcing Data ERA-Interim (WFDEI, Weedon et al., 2014) reanalysis data using meteorological observations in highly polluted Beijing, China (Sect. 3.1), (2) to evaluate the urban land surface model Surface Urban Energy and Water Balance Scheme (SUEWS, Järvi et al., 2011; Ward et al., 2017) using eddy covariance flux measurements of latent and sensible heat for 2006–2009 (Sect 3.2), and (3) to simulate how increased aerosol concentrations modify the local urban hydrological cycle for the period 2001–2013 using SUEWS (Sect. 3.3). Aerosol optical depth observations are used to classify the pollution levels for the assessment of the impact of radiative effect of haze on the local urban water balance in Beijing for 2001–2013. The broader impacts of changes in water balance on local aquatic environment are discussed in Section 4."

17. l. 10-12 "induce" instead "induces" considering the plural "rates". Also the message of the sentence is a bit confused to figure in the abstract. Please simplify your message.
Rephrased as (abstract, l. 11–14):
"Considering all precipitation events, runoff rates are higher during extremely polluted conditions than cleaner conditions, but as the cleanest conditions have high precipitation rates, they induce the largest runoff. Thus, the haze radiative effect is unlikely to modify flash flooding likelihood."

18. l.11-12 this is a justification of the research, isn't it? It should maybe be placed before the general outcomes.
This has been place in the first paragraph and rephrased (page 1, l. 2–4):
"Changes in the hydrological cycle modify surface runoff and flooding. Furthermore, as runoff commonly transports pollutants to soil and water, any changes impact urban soil and aquatic environments."

19. Introduction Page 1, l. 17. Is "northeast China one of the most populated areas" a consequence of environmental problems? Please rephrase.
This is rephrased (from page 1, l. 119 to page 2, l. 1): "As a consequence of urbanization and industrialization, northeast China is one of the most populated and polluted areas in the world"

20. Page 2, l. 1-2. Please rephrase. The sentence is difficult to read in my opinion.

Rephrased as (page 2, l. 2–3):
"With continuing urbanization, serious water shortages and deterioration of aquatic environment are becoming essential concerns for municipal hydrological authorities"

21. Page 2. l 4-5. It can worth it to elaborate more the interactions between the aerosols, the solar radiation, and the boundary layer height and stability. How these elements interact?
A short description has been added to the original text. Now it says (page 2, l. 7–12): "Increased aerosol concentration modifies local urban climate by decreasing solar radiation received at the surface and thus decreasing near surface air temperatures (Ding et al., 2013; Wang et al., 2014), turbulent heat fluxes and boundary layer heights but increases incoming longwave radiation emissions from the polluted atmosphere (Miao et al., 2009; Petäjä et al., 2016; Tang et al., 2016). Increased absorption of radiation by the polluted atmosphere changes the vertical temperature profile leading to more stable conditions (Petäjä, et al., 2016). With reduced turbulence and mixing, the boundary layer height is lower (Petäjä et al, 2016) which increases the near surface pollutant concentrations."

22. Further Page 4 l.10. "as the main focus...balance". This information can be removed and is not so necessary here.
Removed as suggested.

23. Page 4 l.24 Please indicate where are the mentioned box-plots, RMSEs, etc. in the paper?
These have been added to the text in Section 2.1.

24. Page 5 l-9-11. This is more appropriated in the introduction.
This part has been moved to the introduction (page 2, l. 21–24).

25. Page 6 Table 3. Please modify the title to better explain what is presented in this table and which variables are there intercompared.
Rephrased as:
"Comparison for 2006-2009 of WFDEI meteorological variables with observations, stratified by pollution levels (extremely polluted air (AOD>1), polluted air (0.438-1), low pollution (0.203-0.438), fairly clean air (<0.203) (see Sect. 2.1 for details). Data are hourly: relative humidity (RH, %), air temperature (Tair, ºC), and incoming solar radiation (K↓, W m$^{-2}$), and daily: precipitation (P, mm day$^{-1}$). Superscript uc indicates uncorrected variables. For explanation of the statistical methods see Sect. 2.1."

In addition, similar changes have been made for Table 5.

26. Page 11 l.11 –Page 12 l.11. This is an interesting discussion although not a result.
This has been moved to its own section (Section 4. Discussion of broader impacts).

**Referee #2**

The authors evaluated a reanalysis data and a hydrological model with observational data and examined the effects of haze impact on the surface hydrology. The examination of how haze impacts on local scale urban hydrological cycle is particularly interesting. But I have some concerns about the connection between two parts. There also needs some clarifications about the approach and the hydrological model. My specific comments are as below. I recommend a major revision before the paper can be accepted for a publication in ACP.

1. Part 1, Introduction
The authors did not describe the ways how urban pollution can impact on the hydrological cycle.
Short description has been added to the introduction (page 2, l. 12–14): "With less energy available at the surface, because of  the attenuated incoming solar radiation, evaporation may be reduced modifying other water balance terms. In addition, haze can increase the condensation nuclei and therefore precipitation."

2. On Page 2, 2nd paragraph, the authors mentioned the air temperature change by urban pollution, but the focus of the paper is how urban pollution impacts on hydrological cycle. It is necessary to provide some background about how precipitation or water on ground are connected with pollution in atmosphere.
See answer comment #1 above.

3. In the introduction, it needs to be clear that the pollution effects on precipitation are not considered, which would add the uncertainty to the study. It seems to me that the paper only considers the effects induced by changed surface temperature. Pollution can change temperature profile in atmosphere and serve as cloud condensation nuclei to affect precipitation rate and then surface hydrology. But all of these are not considered and the paper needs to be clear about it.
This has been now clarified in the introduction (from page 2, l. 33 to page 3, l. 2).

4. Last paragraph of this section: need to be clear about which method is used for each objective.
Rephrased as (page 2, l. 25–32):
"The aims of this study are (1) to evaluate the 2006 to 2009 WATCH Forcing Data ERA-Interim (WFDEI, Weedon et al., 2014) reanalysis data using meteorological observations in highly polluted Beijing, China (Sect. 3.1), (2) to evaluate the urban land surface model Surface Urban Energy and Water Balance Scheme (SUEWS, Järvi et al., 2011; Ward et al., 2017)  using eddy covariance flux measurements of latent and sensible heat for 2006–2009 (Sect 3.2), and (3) to simulate how increased aerosol concentrations modify the local urban hydrological cycle for the period 2001–2013 using SUEWS (Sect. 3.3). Aerosol optical depth observations are used to classify the pollution levels for the assessment of the impact of radiative effect of haze on the local urban water balance in Beijing for 2001–2013. The broader impacts of changes in water balance on local aquatic environment are discussed in Section 4."

5. Section 2
It is not clear that what are the physical parameters for model input and outputs. Although Table shows some input variables, those are just symbols and their physical meaning is not provided. Most importantly, there is no information about how aerosols/pollution would impact temperature, runoff, and soil infiltration in the model? This is very important to understand what mechanisms are included for pollution impact runoff and soil infiltration.
Air temperature from the reanalysis data is used to force the model. This includes readily the effect of pollutants. This is added to page 6, l. 16. Runoff and soil infiltration are impacted mainly by the changes in the energy balance via the corrected reanalysis attenuated incoming solar radiation data (model forcing). This leads to decreased evaporation rates which modifies the other water balance term. Further explanation is added (from page 2, l. 33 to page 3, l. 1) in addition to the existing text (page 10, l. 13–16). In addition, the effect of haze on the longwave radiation is explained in page 2, l. 9 and page 9, l. 3–5 and shown in Fig. 7. The Table 1 is showing the parameterisation mainly for the reproducing purposes, and it is not showing the relation of the model to the haze. The meaning of each of the variables are shown in Table A1.

6. For the hydrological model, is precipitation rate changed by any factors? Is it just an input, not an output?

The precipitation rate is only an input. It has been clarified in the text that the precipitation is one of the input variables and the changes in precipitation are provided by the reanalysis data (page 2, l. 34; page 3, l. 12; page 5, l. 20 and page 9, l. 2)

7. Section 3
It is not clear to me how section 3.1 and 3.2 are relevant or connected. Or are they just separated results without much connection? Why not run the model validated in Section 3.1 to look at the effect of haze on surface water in Section 3.2? In this way, the connection between the two sections is clear.
Section 3.1 is the evaluation and correction of the WFDEI reanalysis data. Section 3.2 is evaluation of SUEWS model using corrected reanalysis data as meteorological forcing and section 3.3 is the analysis of the effect of haze on the hydrological cycle using SUEWS forced with corrected WFDEI data. Text has been rephrased to clarify this (page 2, l. 25–32).

8. Some symbols are used without being defined in the text, such as P, K,…
The symbols for P and K↓ are defined at page 4, l. 24–25. Also other variables are defined in this page (Tair, RH). The definition of I for irrigation has been changed from page 3, l. 11 to page 10, l. 16–17 for clarity.

9. Section 4
Please use physical terms, not symbols in the conclusion text.
The symbols are now changed to physical terms in the conclusions (Section 5).

[revised manuscript text omitted]

---

## Author Response (AR2)

We thank the reviewers for their comments, these have improved the paper. Our detailed responses are given in blue below.

Page number and line numbers indicated in the responses are the new ones unless otherwise indicated

**Referee #1**

General comments

By evaluating the impact of haze on urban hydrological cycle and limitation of the current modelling multi-scale approach, the paper addresses a relevant scientific issue that will help the scientific communities and decision makers worldwide. This research gains even more in importance in highly dense Asian megacities (China, India) that already suffer from aerosol pollution. By talking modelling approach and atmospheric chemical and physical processes, the reviewer agrees that the topics entirely fit with the concerns of the Atmospheric Chemistry and Physics journal. In general the manuscript is well written but effort can be make to simplify the sentences (some-time confusing) and on the abstract/introduction to clarify the aims (as it seems there are several) and take-out messages of the research. It seems that the paper questions the modelling approach and particularly the quality of the global reanalysis data used to simulate the local urban hydrological cycles during haze episodes, the SUEWS urban land surface model performance as well as the interactions between the aerosols and the urban hydrological. This should be clearly stated from the abstract until the result sections. The novelty (the focus on the local scale), challenges, and operational urban water management implication raised in the paper should also be better justified in the introduction and generic terms should be avoid to go directly to the fact (aerosols instead of pollution) and determinant physical interactions treated by the papers. The introduction can be elaborated so as to immediately focus the reader on the nature of the pollution the authors are dealing with (aerosols and wet haze?) instead to use generic terms. It will help to strengthen the message of the introduction. Following are specific and some technical comments/corrections that will hopefully be helpful to the authors.

*Response*
*Abstract changes*
The justification of the study is moved from the end of the abstract to the first paragraph and rephrased (page 1, l. 2–4): "Changes in hydrological cycle modify surface runoff and flooding. Furthermore, as runoff commonly transports pollutants to soil and water, any changes impact urban soil and aquatic environments."

Also the aims of the study (evaluation of the model and reanalysis data and the simulation of hydrological cycle) are rephrased to clarify it.

*Introduction changes*
The introduction text is rephrased to raise the novelty of local scale and the potential deterioration of urban water bodies (page 2, l. 14–17): "The higher surface runoff rates due to the modified water balance may increase pollutant loads in urban water bodies, by flushing pollutants from contaminated surfaces. However, the linkage between increased aerosol concentrations and urban hydrological cycle has not yet been studied in local scale despite its potential contribution to deterioration of urban aquatic environment."

The atmospheric (or air) pollution is rephrased in introduction to increased atmospheric aerosol concentration (page 1, l. 18, page 2, l. 7 and 16).

Description of physical interactions of pollutant concentrations, solar radiation, boundary layer height and its effects on water balance is added to the introduction (page 2, l. 10–14).

A short description of the importance of the accuracy of the reanalysis data is added (page 2, l. 21–24).

Specific comments

1. Page 3, l.8. Is it possible to explain the specificity of the Murto(2017)'s methods. Murto (2017) does not detail enough the method behind the landcover model construction and how the various vegetation compositions are retrieved from the aerial photographs.
The methods are widely used methods with ArcGIS-software using OpenStreetMap (OSM) and World Imagery, as described in the text (page 4, l. 2–7), and not specific to this study. With the available imagery the vegetation composition and tree height is not possible to obtain with GIS-methods. Therefore, it is estimated from the common tree species in Haidian district as described in page 4, l. 7–10.

2. What is the benefit of using the Murto's method and two source of spatial information?
There are substantial gaps in the OSM data in the region, especially at the southern part of the study area. Therefore two sources were used to evaluate the existing data at the areas where the two sources could be compared. This information is added to the text (page 4, l. 5–7).

3. What is the resolution of the World imagery?
The spatial resolution is 1m. This has been added to the text (page 4, l. 4).

4. Why a semi supervised classification was not able to distinguished evergreen from deciduous trees based on irradiance and trees from shrubs based on a structural geometry algorithm?
It might have been possible to distinguish the evergreen and deciduous trees from irradiance data, but from the data used this was not possible. However, the method used, where the classification is made using the studied proportions of the trees in the district should give high enough accuracy.

The fraction of trees and shrubs are given to the model as one input variable where both are included (for evergreen and deciduous separately) so there is no need to identify trees from shrubs.

5. What is the quality of the OSM data in the region?
See, comment #2 above.

6. Page 4 Can you confirm that the model has been run for a time period of 3 years and a 5 min time step over a 1km2 simulation domain?
The evaluation of the WFDEI data and the SUEWS model has been made for a time period of 4 years (2006–2009), since it is the time when observations were available. This is stated at page 2, l. 25 and 28 and page 4, l. 23–26. For the analysis of the hydrological cycle, the model is run for the period of 14 years (2000–2013), where the first year is used as a spin-up period, leaving years 2001–2013 for the analysis (page 4, l. 29). The time step for the model runs is 5 min. The study area is a 1 km radius circle around the IAP tower (page 3, l. 17–18) (i.e. area= 3.14 $km^2$).

7. Page 5 l.28. What was the nMBE before the correction?
The nMBE before correction (-0.01, 0.00, 0.08, 0.28 from clean to extremely polluted conditions) were added to the text (page 6, l. 13).

8. Page 8, l.1. "SUEWS model performance is relatively independent of haze level (...) in the model input variables". Is the precipitation not also an input variable affected by the haze levels?
Precipitation is added to the variables affected by the haze (page 9, l. 3).

9. How the bias in the precipitation can impact the quality of the simulation with respect to the precipitation rates (p.8 l5.)?
The surface runoff is the most sensitive to precipitation, but it is analysed using normalized values and therefore the uncertainties in precipitation are not crucial to the conclusions (page 9, l. 7–8). Also Fig. 5 and Table 5 show that the model performance is not substantially decreasing with increasing air pollution.

10. What about the influence of haze level on the longwave radiations, surface temperature and resulting QH and atmosphere stability?
The effect of haze on longwave radiation is shown in Fig. 7 and it is discussed in the text (page 9, l. 3–5 and page 2, l. 9).

The evaluation against the sensible heat flux which is also related to surface temperature is included in Fig. 5 and Table 5 showing the statistics. It is discussed in the text (page 10, l. 1–5 and page 14, l. 4–6).

The further analysis of QH and the stability of the atmosphere is out of the scope of this paper, as we focus on the hydrological cycle and surface processes. However, a short description of the effect of aerosols on atmospheric stability is included in page 2, l. 10–11 and page 10, l. 11–12.

11. Should the model performance only be evaluated with the evaporation? Evapotranspiration is the common term in the energy and precipitation budget but as the incoming energy is partitioned also amongst other terms (sensible, storage), does it worth it to also consider these variable in the evaluation of the simulations.
The model evaluation against sensible heat flux is included (see above comment #10). Unfortunately, the evaluation against storage heat flux is not possible due to difficulty of measuring this flux.

12. A fortiori, aerosols have been proven to increase the contributions of the scattered radiation versus direct radiation in the solar energy budget,while they potentially absorb and emit longwave radiation resulting in heat retention in the atmosphere.
Short description of this has been added (see above comment #10)

13. Table 2. What are the uncertainties associated with the temperature, humidity and wind speed sensors developed by the institute of Atmospheric Physics? As being nonstandard instruments, is it possible to have a description of these and know if they have been already tested against standard sensors?
We do not have detailed description of these instruments, but these have been widely used in previous internationally peer reviewed papers (e.g., Liu et al., (2012) Atmos. Chem. Phys. 12, 7881–7892; Song and Wang, (2012) Atmos. Res. 106, 139–149; Shi et al., (2018) Atmospheric and Oceanic Science Letters 12, 41–49; Al-Jiboori and Hu, (2005) Adv. Atmos. Sci. 22, 595–605)

14. Page 10 l.11. Is surface runoff not diminished for small precipitation intensities compared to high precipitation intensity episodes? The infiltration capacity of the soil horizon is usually reduced during high precipitation intensity episodes due to the destruction of soil aggregates -> less porosity, and usually deeper wetting front in the soil resulting in higher surface resistance. Please clarify.

Yes the surface runoff is smaller with small daily precipitation totals (as also seen in Fig. 4 and A2) for the above-mentioned reasons. However, surface runoff does not totally vanished as there are a lot of impervious surfaces (buildings and paved surfaces) that still generate surface runoff.

Technical corrections

15. Abstract l. 5-6 please, rephrase.
"We show how the reanalysis radiation data do not include the attenuating effect of haze and develop a haze correction for the incoming solar radiation. With this haze correction the SUEWS model simulates the eddy covariance measured latent heat flux well."
Rephrased as (abstract, l. 8–9):
"We show that the reanalysis data do not include the attenuating effect of haze on incoming solar radiation and develop a correction method. Using these corrected data, SUEWS simulates measured eddy covariance heat fluxes well."

16. Additionally, it was not clearly stated before that the evaluation of the SUEWS performance is also part of the aim of the paper. It can be good either to neglect this aspect in the abstract or if crucial add this additional and somehow "hidden" aim in the abstract.
In abstract it is rephrased (l. 6–7): "The secondary aims are to examine the usability of global reanalysis dataset in highly polluted environment and the SUEWS model performance."
In addition the introduction is rephrased as (page 2, l. 25–32):
"The aims of this study are (1) to evaluate the 2006 to 2009 WATCH Forcing Data ERA-Interim (WFDEI, Weedon et al., 2014) reanalysis data using meteorological observations in highly polluted Beijing, China (Sect. 3.1), (2) to evaluate the urban land surface model Surface Urban Energy and Water Balance Scheme (SUEWS, Järvi et al., 2011; Ward et al., 2017) using eddy covariance flux measurements of latent and sensible heat for 2006–2009 (Sect 3.2), and (3) to simulate how increased aerosol concentrations modify the local urban hydrological cycle for the period 2001–2013 using SUEWS (Sect. 3.3). Aerosol optical depth observations are used to classify the pollution levels for the assessment of the impact of radiative effect of haze on the local urban water balance in Beijing for 2001–2013. The broader impacts of changes in water balance on local aquatic environment are discussed in Section 4."

17. l. 10-12 "induce" instead"induces" considering the plural "rates". Also the message of the sentence is a bit confused to figure in the abstract. Please simplify your message.
Rephrased as (abstract, l. 11–14):
"Considering all precipitation events, runoff rates are higher during extremely polluted conditions than cleaner conditions, but as the cleanest conditions have high precipitation rates, they induce the largest runoff. Thus, the haze radiative effect is unlikely to modify flash flooding likelihood."

18. l.11-12 this is a justification of the research, isn't it? It should maybe be placed before the general outcomes.
This has been place in the first paragraph and rephrased (page 1, l. 2–4):
"Changes in the hydrological cycle modify surface runoff and flooding. Furthermore, as runoff commonly transports pollutants to soil and water, any changes impact urban soil and aquatic environments."

19. Introduction Page 1, l. 17. Is "northeast China one of the most populated areas" a consequence of environmental problems? Please rephrase.
This is rephrased (from page 1, l. 119 to page 2, l. 1): "As a consequence of urbanization and industrialization, northeast China is one of the most populated and polluted areas in the world"

20. Page 2, l. 1-2. Please rephrase. The sentence is difficult to read in my opinion.

Rephrased as (page 2, l. 2–3):
"With continuing urbanization, serious water shortages and deterioration of aquatic environment are becoming essential concerns for municipal hydrological authorities"

21. Page 2. l 4-5. It can worth it to elaborate more the interactions between the aerosols, the solar radiation, and the boundary layer height and stability. How these elements interact?
A short description has been added to the original text. Now it says (page 2, l. 7–12): "Increased aerosol concentration modifies local urban climate by decreasing solar radiation received at the surface and thus decreasing near surface air temperatures (Ding et al., 2013; Wang et al., 2014), turbulent heat fluxes and boundary layer heights but increases incoming longwave radiation emissions from the polluted atmosphere (Miao et al., 2009; Petäjä et al., 2016; Tang et al., 2016). Increased absorption of  radiation by the polluted atmosphere changes the vertical temperature profile leading to more  stable conditions (Petäjä, et al., 2016). With reduced turbulence and mixing, the boundary layer height is lower (Petäjä et al, 2016) which increases the near surface pollutant concentrations."

22. Further Page 4 l.10. "as the main focus...balance". This information can be removed and is not so necessary here.
Removed as suggested.

23. Page 4 l.24 Please indicate where are the mentioned box-plots, RMSEs, etc. in the paper?
These have been added to the text in Section 2.1.

24. Page 5 l-9-11. This is more appropriated in the introduction.
This part has been moved to the introduction (page 2, l. 21–24).

25. Page 6 Table 3. Please modify the title to better explain what is presented in this table and which variables are there intercompared.
Rephrased as:
"Comparison for 2006-2009 of WFDEI meteorological variables with observations, stratified by pollution levels (extremely polluted air (AOD>1), polluted air (0.438-1), low pollution (0.203-0.438), fairly clean air (<0.203) (see Sect. 2.1 for details). Data are hourly: relative humidity (RH, %), air temperature (Tair, ºC), and incoming solar radiation (K↓, W m$^{-2}$), and daily: precipitation (P, mm day$^{-1}$). Superscript uc indicates uncorrected variables. For explanation of the statistical methods see Sect. 2.1."

In addition, similar changes have been made for Table 5.

26. Page 11 l.11 –Page 12 l.11. This is an interesting discussion although not a result.
This has been moved to its own section (Section 4. Discussion of broader impacts).

**Referee #2**

The authors evaluated a reanalysis data and a hydrological model with observational data and examined the effects of haze impact on the surface hydrology. The examination of how haze impacts on local scale urban hydrological cycle is particularly interesting. But I have some concerns about the connection between two parts. There also needs some clarifications about the approach and the hydrological model. My specific comments are as below. I recommend a major revision before the paper can be accepted for a publication in ACP.

1. Part 1, Introduction
The authors did not describe the ways how urban pollution can impact on the hydrological cycle.
Short description has been added to the introduction (page 2, l. 12–14): "With less energy available at the surface, because of the attenuated incoming solar radiation, evaporation may be reduced modifying other water balance terms. In addition, haze can increase the condensation nuclei and therefore precipitation."

2. On Page 2, 2nd paragraph, the authors mentioned the air temperature change by urban pollution, but the focus of the paper is how urban pollution impacts on hydrological cycle. It is necessary to provide some background about how precipitation or water on ground are connected with pollution in atmosphere.
See answer comment #1 above.

3. In the introduction, it needs to be clear that the pollution effects on precipitation are not considered, which would add the uncertainty to the study. It seems to me that the paper only considers the effects induced by changed surface temperature. Pollution can change temperature profile in atmosphere and serve as cloud condensation nuclei to affect precipitation rate and then surface hydrology. But all of these are not considered and the paper needs to be clear about it.
This has been now clarified in the introduction (from page 2, l. 33 to page 3, l. 2).

4. Last paragraph of this section: need to be clear about which method is used for each objective.
Rephrased as (page 2, l. 25–32):
"The aims of this study are (1) to evaluate the 2006 to 2009 WATCH Forcing Data ERA-Interim (WFDEI, Weedon et al., 2014) reanalysis data using meteorological observations in highly polluted Beijing, China (Sect. 3.1), (2) to evaluate the urban land surface model Surface Urban Energy and Water Balance Scheme (SUEWS, Järvi et al., 2011; Ward et al., 2017) using eddy covariance flux measurements of latent and sensible heat for 2006–2009 (Sect 3.2), and (3) to simulate how increased aerosol concentrations modify the local urban hydrological cycle for the period 2001–2013 using SUEWS (Sect. 3.3). Aerosol optical depth observations are used to classify the pollution levels for the assessment of the impact of radiative effect of haze on the local urban water balance in Beijing for 2001–2013. The broader impacts of changes in water balance on local aquatic environment are discussed in Section 4."

5. Section 2
It is not clear that what are the physical parameters for model input and outputs. Although Table shows some input variables, those are just symbols and their physical meaning is not provided. Most importantly, there is no information about how aerosols/pollution would impact temperature, runoff, and soil infiltration in the model? This is very important to understand what mechanisms are included for pollution impact runoff and soil infiltration.
Air temperature from the reanalysis data is used to force the model. This includes readily the effect of pollutants. This is added to page 6, l. 16. Runoff and soil infiltration are impacted mainly by the changes in the energy balance via the corrected reanalysis attenuated incoming solar radiation data (model forcing). This leads to decreased evaporation rates which modifies the other water balance term. Further explanation is added (from page 2, l. 33 to page 3, l. 1) in addition to the existing text (page 10, l. 13–16). In addition, the effect of haze on the longwave radiation is explained in page 2, l. 9 and page 9, l. 3–5 and shown in Fig. 7. The Table 1 is showing the parameterisation mainly for the reproducing purposes, and it is not showing the relation of the model to the haze. The meaning of each of the variables are shown in Table A1.

6. For the hydrological model, is precipitation rate changed by any factors? Is it just an input, not an output?

The precipitation rate is only an input. It has been clarified in the text that the precipitation is one of the input variables and the changes in precipitation are provided by the reanalysis data (page 2, l. 34; page 3, l. 12; page 5, l. 20 and page 9, l. 2)

7. Section 3
It is not clear to me how section 3.1 and 3.2 are relevant or connected. Or are they just separated results without much connection? Why not run the model validated in Section 3.1 to look at the effect of haze on surface water in Section 3.2? In this way, the connection between the two sections is clear.
Section 3.1 is the evaluation and correction of the WFDEI reanalysis data. Section 3.2 is evaluation of SUEWS model using corrected reanalysis data as meteorological forcing and section 3.3 is the analysis of the effect of haze on the hydrological cycle using SUEWS forced with corrected WFDEI data. Text has been rephrased to clarify this (page 2, l. 25–32).

8. Some symbols are used without being defined in the text, such as P, K,…
The symbols for P and K↓ are defined at page 4, l. 24–25. Also other variables are defined in this page (Tair, RH). The definition of I for irrigation has been changed from page 3, l. 11 to page 10, l. 16–17 for clarity.

9. Section 4
Please use physical terms, not symbols in the conclusion text.
The symbols are now changed to physical terms in the conclusions (Section 5).

**Simulation of the radiative effect of haze on urban hydrological cycle using reanalysis data in Beijing**

Tom V. Kokkonen[1], Sue Grimmond[2], Sonja Murto[1,3], Huizhi Liu[4], Anu-Maija Sundström[5], and Leena Järvi[1,6]

[1]Institute for Atmospheric and Earth System Research / Physics, Faculty of Science, University of Helsinki, Finland
[2]Department of Meteorology, University of Reading, UK
[3]Department of Meteorology, University of Stockholm, Sweden
[4]Institute of Atmospheric Physics, Chinese Academy of Sciences, Beijing, China
[5]Earth Observation, Finnish Meteorological Institute, Helsinki, Finland
[6]Helsinki Institute of Sustainability Science, University of Helsinki, Finland

**Correspondence:** Tom Kokkonen (tom.kokkonen@helsinki.fi), Huizhi Liu (huizhil@mail.iap.ac.cn)

**Abstract.** Although, air pollution increased aerosol concentration modifies local air temperatures and boundary layer structure in urban areas, little is known about its effects on the urban hydrological cycle. Changes in the hydrological cycle modify surface runoff and flooding. Furthermore, as runoff commonly transports pollutants to soil and water, any changes impact urban soil and aquatic environments. To explore the radiative effect of haze on changes in the urban surface water balance during in Beijing, different haze levels are modelled in Beijing using the Surface Urban Energy and Water Balance Scheme (SUEWS), forced by reanalysis data. The pollution levels are classified using aerosol optical depth observations. We show how the reanalysis radiation The secondary aims are to examine the usability of global reanalysis dataset in highly polluted environment and the SUEWS model performance.

We show that the reanalysis data do not include the attenuating effect of haze on incoming solar radiation and develop a haze correction for the incoming solar radiation. With this haze correction the SUEWS model simulates the eddy covariance measured latent heat flux well. correction method. Using these corrected data, SUEWS simulates measured eddy covariance heat fluxes well. Both surface runoff and drainage increase with severe haze levels particularly with low precipitation rates: runoff from 0.06 to 0.18 mm day$^{-1}$ and drainage from 0.43 to 0.62 mm day$^{-1}$ during fairly clean and to extremely polluted conditions, respectively. When Considering all precipitation events are taken into account, runoff is higher during the , runoff rates are higher during extremely polluted conditions than with cleaner conditions except during cleaner conditions, but as the cleanest conditions when the have high precipitation rates induces , they induce the 
[revised manuscript text omitted]